# Synthesizing a Progression of Subtasks
# for Block-Based Visual Programming Tasks

## Abstract

Block-based visual programming environments play an increasingly important role in introducing computing concepts to K-12 students. In recent years, they have also gained popularity in neuro-symbolic AI, serving as a benchmark to evaluate general problem-solving and logical reasoning skills. The open-ended and conceptual nature of these visual programming tasks make them challenging, both for state-of-the-art AI agents as well as for novice programmers. A natural approach to providing assistance for problem-solving is breaking down a complex task into a progression of simpler subtasks; however, this is not trivial given that the solution codes are typically nested and have non-linear execution behavior. In this paper, we formalize the problem of synthesizing such a progression for a given reference block-based visual programming task. We propose a novel synthesis algorithm that generates a progression of subtasks that are high-quality, well-spaced in terms of their complexity, and solving this progression leads to solving the reference task. We show the utility of our synthesis algorithm in improving the efficacy of AI agents (in this case, neural program synthesizers and search-based agents) for solving tasks in the *Karel programming environment* (Pattis et al., 1995). Then, we conduct a user study to demonstrate that our synthesized progression of subtasks can assist a novice programmer in solving tasks in the *Hour of Code: Maze Challenge* (Code.org, 2022c) by *Code.org* (Code.org, 2022a).

## 1 Introduction

The emergence of block-based visual programming platforms has made coding more accessible and appealing to novice programmers, including K-12 students. Led by the success of programming environments like *Scratch* (Resnick et al., 2009) and *Karel* (Pattis et al., 1995), initiatives like *Hour of Code* (Code.org, 2022b) by *Code.org* (Code.org, 2022a) and online platforms like *CodeHS.com* (CodeHS, 2022), block-based programming has become an integral part of introductory computer science education. Importantly, in contrast to typical text-based programming, block-based visual programming reduces the burden of learning syntax and puts direct emphasis on fostering computational thinking and general problem-solving (Weintrop & Wilensky, 2015; Price & Barnes, 2017; 2015). This unique aspect, in turn, also makes block-based visual programming environments an interesting benchmark for neuro-symbolic AI, in particular, to evaluate agents' problem-solving and logical reasoning skills (Schuster et al., 2021; Puri et al., 2021; Li et al., 2022).

Programming tasks on these platforms are open-ended and require multi-step deductive reasoning to solve. As a result, novices may struggle when solving these tasks, leading to low success rate in finding a correct solution (Piech et al., 2015; Ghosh et al., 2022; Price et al., 2017; Wu et al., 2019). Similarly, solving these tasks can also be challenging for state-of-the-art AI agents (neural program synthesizers) trained using reinforcement learning methods; as observed in Bunel et al. (2018), the agents' performance decreased drastically with increased nesting in the solution codes. A natural way to deal with task complexity is to "break down a task" into subtasks—this framework of subtasking has proven to be effective in several domains (Decker et al., 2019; Morrison et al., 2015), including geometry proof problems (McKendree, 1990), Parson's coding problems (Morrison et al., 2016), and robotics (Bakker & Schmidhuber, 2004; Ding et al., 2014). Inspired by this, we seek to develop algorithms that can synthesize a progression of subtasks for block-based visual programming tasks.

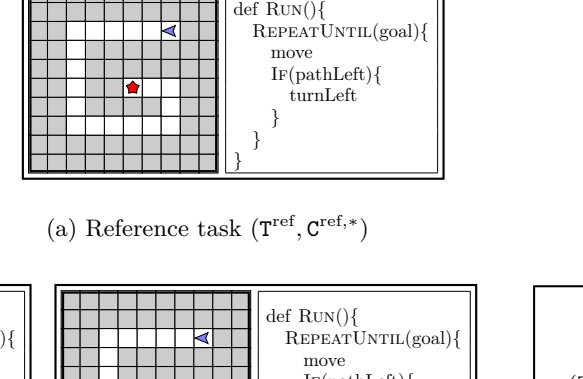

(a) Reference task $(\mathtt{T}^{\mathrm{ref}}, \mathtt{C}^{\mathrm{ref},*})$

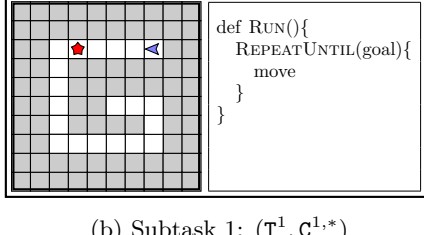

(b) Subtask 1: $(\mathtt{T}^1, \mathtt{C}^{1,*})$

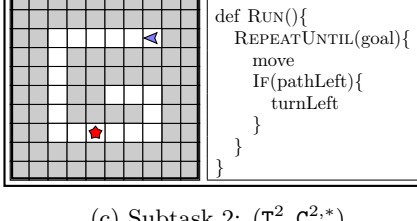

(c) Subtask 2: $(\mathtt{T}^2, \mathtt{C}^{2,*})$

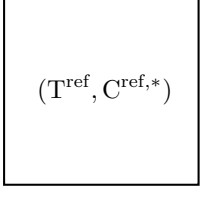

(d) Subtask 3: $(\mathtt{T}^3, \mathtt{C}^{3,*})$

Figure 1: Illustration of our synthesis algorithm on Maze16 task from the *Hour of Code: Maze Challenge* Code.org (2022c) by *Code.org* Code.org (2022a). **(a)** shows visual grid $\mathtt{T}^{\mathrm{ref}}_{\mathrm{vis}}$ of reference task $\mathtt{T}^{\mathrm{ref}}$ and its solution code $\mathtt{C}^{\mathrm{ref},*}$ which are provided as input to our synthesis algorithm. When solving this task, one needs to write a code that upon execution would navigate the "avatar" (purple dart) to the "goal" (red star) in the visual grid. Additionally, the maximum number of allowed code blocks in the solution code is $\mathtt{T}^{\mathrm{ref}}_{\mathrm{size}} = 4$ and allowed types of blocks are $\mathtt{T}^{\mathrm{ref}}_{\mathrm{store}} = \{\textsc{RepeatUntil}, \textsc{If}, \mathtt{move}, \mathtt{turnLeft}, \mathtt{turnRight}\}$. **(b)**, **(c)**, and **(d)** show the progression of $K = 3$ subtasks for the reference task synthesized by our algorithm PROGRESSYN. As can be seen, the subtasks are well-spaced w.r.t. their complexity and the visual grids are minimal modifications of the visual grid $\mathtt{T}^{\mathrm{ref}}_{\mathrm{vis}}$.

However, automatically synthesizing such a progression for block-based visual programming domains is non-trivial. Codes have nested structures, their execution behaviour on task's visual grid is "non-linear"; hence, existing subtasking techniques (in domains such as path-navigation or robotics) relying on well-defined "linear" behaviors do not apply to our setting (Bakker & Schmidhuber, 2004; Ding et al., 2014). Also, the space of visual grids and codes is potentially unbounded, and the mapping between these spaces is highly discontinuous (i.e., a small modification can make the task invalid); hence, techniques relying on exhaustive enumeration to find valid/high-quality subtasks are intractable (Ahmed et al., 2020; Polozov et al., 2015).

## 1.1 Our Approach and Main Contributions

We begin by formalizing our objective of synthesizing a progression of programming subtasks. Our proposed synthesis methodology overcomes the key challenges discussed above by reasoning about the execution behavior of the provided solution code on the reference task's visual grids. As concrete examples, consider reference tasks in Figures 1a and 2a from two popular platforms. Given such a reference task along with a solution code as input, we seek to synthesize a progression of subtasks with the following properties: (a) each subtask is a high-quality programming task by itself; (b) the complexity of solving a subtask in the progression increases gradually, i.e., the subtasks are well-spaced w.r.t. their complexity; (c) solving the progression would help in increasing success rate of solving the reference task.

Our main contributions and results are as follows: (I) We formalize the objective of synthesizing a progression of subtasks for block-based visual programming tasks and present a novel synthesis algorithm, PROGRESSYN (Sections 2 and 3); (II) We showcase the utility of PROGRESSYN towards assisting the problem-solving process of AI agents via simulations and of novice programmers via an online study (Sections 4 and 5); (III) We will

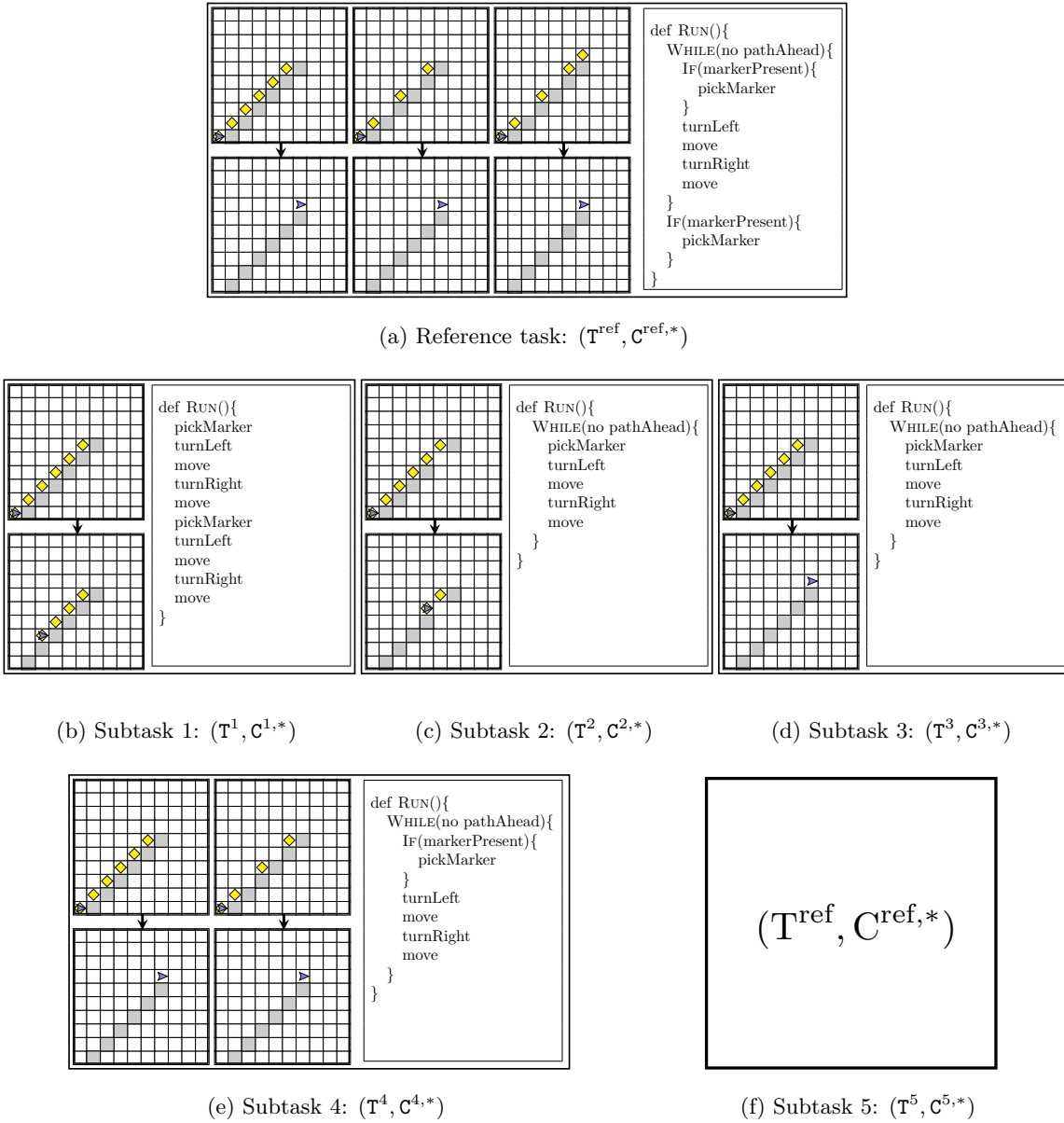

Figure 2: Illustration of our synthesis algorithm on an adaptation of STAIRWAY task from *CodeHS.com* CodeHS (2022) based on *Karel programming environment* Pattis et al. (1995). Compared to the task shown in Figure 1, Karel tasks are more complex and include multiple I/O pairs. **(a)** shows the three I/O pairs of reference task $T^{ref}$ and its solution code $C^{ref,*}$, provided as input to our synthesis algorithm. When solving the task, the goal is to write code that upon execution, converts pregrid (upper grid of a visual I/O pair) to its corresponding postgrid (lower grid of a visual I/O pair, and connected to the pregrid by an arrow) for all I/O pairs. The visual grids additionally have yellow diamond "markers" which can be put or picked from a grid cell using the `putMarker` and `pickMarker` code blocks respectively. **(b–f)** show the progression of $K = 5$ subtasks for the reference task synthesized by our algorithm PROGRESSYN. In particular, the subtasks in (b-d) are synthesized from the first I/O pair in (a) and the subtasks in (e-f) gradually bring in the other two I/O pairs of (a).

publicly share the web app used in the study and the implementation of PROGRESSYN to facilitate future research (Appendix C.4, Appendix F.1, and supplementary material).[1]

---

[1]While our proposed algorithm is designed for block-based visual programming environments, it could potentially be extended to other domains (such as text-based programming environments) if the following elements are redesigned: (a) task synthesis for programs in the domain using symbolic execution; (b) task complexity; (c) task dissimilarity.

## 1.2 Related Work

**Subtasks in programming education.** Prior work in developing subtasks for block-based programming tasks requires access to resources such as expert labels or students' historical data on these platforms (Margulieux et al., 2019; 2020; Marwan et al., 2021). For instance, Marwan et al. (2021) uses unsupervised clustering techniques to automatically detect common code patterns in student data, followed by hierarchical clustering methods to combine frequently co-occurring code patterns into subtasks. Our approach is different in that we seek to automate the generation of subtasks without access to any prior data.

**Neural program synthesis (NPS).** In recent years, a number of neural models have been proposed which learn to synthesize solution codes for text-based and visual block-based programming tasks (Balog et al., 2017; Austin et al., 2021; Devlin et al., 2017; Chen et al., 2019). However, these models are data hungry, which has led to a significant amount of work in creating synthetic datasets for training these models (Ahmed et al., 2020; Laich et al., 2020; Bunel et al., 2018; Suh & Timen, 2020). PROGRESSYN complements this line of work by decomposing existing tasks into subtasks – these subtasks when augmented with the original dataset would increase its diversity further. We validate this property of our synthesized subtasks in Section 4.1.

## 2 Problem Setup

In this section, we introduce definitions and formalize our objective. We have provided a detailed table of notations in Appendix B.

### 2.1 Preliminaries

**The space of tasks and codes.** We define a task $\mathtt{T}$ as a tuple $(\mathtt{T_n}, \mathtt{T_{vis}}, \mathtt{T_{store}}, \mathtt{T_{size}})$ where $\mathtt{T_n}$ denotes the number of visual grids, $\mathtt{T_{vis}} := \{\mathtt{T_{vis},i}\}_{i=1,\ldots,\mathtt{T_n}}$ denotes the set of $\mathtt{T_n}$ visual grids, $\mathtt{T_{store}}$ denotes the types of code blocks available, and $\mathtt{T_{size}}$ denotes the maximum number of code blocks allowed in the solution code. We denote the space of tasks as $\mathbb{T}$. We also define a few important properties of tasks in this space through functions that can be instantiated as desired: (i) $\mathcal{F}^{\mathbb{T}}_{\mathrm{complex}} : \mathbb{T} \to \mathbb{R}$ measures the complexity of solving a task; (ii) $\mathcal{F}^{\mathbb{T}}_{\mathrm{qual}} : \mathbb{T} \to \mathbb{R}$ measures the general quality of a task (Ahmed et al., 2020); (iii) $\mathcal{F}^{\mathbb{T}}_{\mathrm{diss}} : \mathbb{T} \times \mathbb{T} \to \mathbb{R}$ captures the dissimilarity between any two tasks in $\mathbb{T}$. For instance, given the visual nature of the tasks, a possible dissimilarity metric can be the hamming-distance between their visual grids (Norouzi et al., 2012; Ahmed et al., 2020). The domain specific language (DSL) of the programming environment defines the space of codes $\mathbb{C}$ (Ahmed et al., 2020; Bunel et al., 2018)). A code $\mathtt{C} \in \mathbb{C}$ is characterized by the following attributes: $\mathtt{C_{depth}}$ measures the depth of the abstract syntax tree (AST) of $\mathtt{C}$, $\mathtt{C_{size}}$ is the number of code blocks in $\mathtt{C}$, and $\mathtt{C_{blocks}}$ is the types of code blocks in $\mathtt{C}$. We also define a code complexity measure using function $\mathcal{F}^{\mathbb{C}}_{\mathrm{complex}}$. Typically, in block-based programming environments, complexity of a code depends on the depth and size of its AST (Piech et al., 2015; Ghosh et al., 2022). Motivated by this, we define $\mathcal{F}^{\mathbb{C}}_{\mathrm{complex}} = \kappa * \mathtt{C_{depth}} + \mathtt{C_{size}}$, where $\kappa \in \mathbb{N}$. For instance, empty code $\{\textsc{Run}\}$ has complexity $\kappa*1+0$, and code $\{\textsc{Run}\{\textsc{Repeat}(4)\{\texttt{move}\}\}\}$ has complexity $\kappa*2+2$.

**Solution code of a task.** We define $\mathtt{C} \in \mathbb{C}$ as a solution code for a task $\mathtt{T} \in \mathbb{T}$ if all of the following conditions hold: execution of $\mathtt{C}$ solves all the $\mathtt{T_n}$ visual grids of $\mathtt{T}$, $\mathtt{C_{size}} \leq \mathtt{T_{size}}$, and $\mathtt{C_{blocks}} \subseteq \mathtt{T_{store}}$. We denote a specific solution code of a task $\mathtt{T}$ as $\mathtt{C^{T,*}}$; for a task $\mathtt{T^{id}}$, we denote its solution code as $\mathtt{C^{id,*}}$ for brevity. We denote $\mathbb{C}_\mathtt{T}$ as the set of all solution codes of $\mathtt{T}$. Using the notion of solution code of a task, we can specify our complexity measure of a task in this domain. In particular, we define $\mathcal{F}^{\mathbb{T}}_{\mathrm{complex}}(\mathtt{T}) = \min_{\mathtt{C} \in \mathbb{C}_\mathtt{T}} \mathcal{F}^{\mathbb{C}}_{\mathrm{complex}}(\mathtt{C})$.[2]

### 2.2 Objective

Our goal is to synthesize a progression of subtasks for a given reference task, such that solving this progression increases the success rate of solving the reference task. We use the increased success rate as a proxy for measuring the helpfulness of our synthesized progression. Next, we introduce the notion of progression of subtasks and its complexity, and then formalize our synthesis objective.

---

[2]In block-based visual programming domains considered here, the set of solution codes for a given reference task $\mathbb{C}_\mathtt{T}$ is typically small in size. However, if this is not the case, we can define $\mathbb{C}_\mathtt{T}$ as a small set of representative solution codes for the reference task.

**Progression of subtasks.** For a reference task $\mathtt{T}^{\text{ref}}$, its solution code $\mathtt{C}^{\text{ref},*}$, and a fixed budget $K$, we denote a progression of subtasks for $\mathtt{T}^{\text{ref}}$ as a sequence $\omega(\mathtt{T}^{\text{ref}}, \mathtt{C}^{\text{ref},*}, K) := ((\mathtt{T}^k, \mathtt{C}^{k,*}))_{k=1,2,\dots,K}$ where the following holds $\forall k$: (a) $\mathtt{C}^{k,*}$ is the solution code of $\mathtt{T}^k$; (b) $\mathtt{T}_\text{n}^k \leq \mathtt{T}_\text{n}^{\text{ref}}$; (c) $\mathtt{T}_\text{store}^k \subseteq \mathtt{T}_\text{store}^{\text{ref}}$. We also have $\mathtt{T}^K = \mathtt{T}^{\text{ref}}$ and $\mathtt{C}^{K,*} = \mathtt{C}^{\text{ref},*}$. We denote the set of all such progressions of $K$-subtasks as $\Omega(\mathtt{T}^{\text{ref}}, \mathtt{C}^{\text{ref},*}, K)$.

**Complexity of a progression of subtasks.** We capture the complexity of a progression of subtasks using the function $\mathcal{F}_{\text{complex}}^\Omega$. Specifically, $\mathcal{F}_{\text{complex}}^\Omega(\omega; \mathtt{T}^{\text{ref}}, \mathtt{C}^{\text{ref},*}, K)$ for a given reference task $\mathtt{T}^{\text{ref}}$ captures the worst case complexity jump in the solution codes of subtasks of $\omega$. More formally:

$$\mathcal{F}_{\text{complex}}^\Omega(\omega; \mathtt{T}^{\text{ref}}, \mathtt{C}^{\text{ref},*}, K) = \max_{k \in \{1,\dots,K\}} \left\{ \min_{k' \in \{0,\dots,k-1\}} \left\{ \mathcal{F}_{\text{complex}}^\mathbb{C}(\mathtt{C}^{k,*}) - \mathcal{F}_{\text{complex}}^\mathbb{C}(\mathtt{C}^{k',*}) \right\} \right\} \tag{1}$$

where $\mathtt{C}^{k,*}/\mathtt{C}^{k',*}$ denote solution codes of subtasks $k/k'$ in $\omega$, and $\mathtt{C}^{0,*}$ denotes empty code $\{\texttt{RUN}\}$.

**Our synthesis objective.** Our objective is to synthesize a progression of $K$ subtasks for a given reference task $\mathtt{T}^{\text{ref}}$ with minimal complexity w.r.t. Equation 1. Our formalism is based on the intuition that lowering complexity reduces the cognitive load of solving the progression, while still being helpful in assisting problem-solving of the reference task (McKendree, 1990). More formally, we seek to generate a progression of subtasks based on the following:

$$\text{Minimize}_{\omega \in \Omega(\mathtt{T}^{\text{ref}}, \mathtt{C}^{\text{ref},*}, K)} \mathcal{F}_{\text{complex}}^\Omega(\omega; \mathtt{T}^{\text{ref}}, \mathtt{C}^{\text{ref},*}, K). \tag{2}$$

In addition, we seek to have the following three desirable properties in our synthesized progression of subtasks. First, subtasks should be of high quality, i.e., maximize $\Sigma_{k \in \{1,\dots,K\}} \mathcal{F}_{\text{qual}}^\mathbb{T}(\mathtt{T}^k)$. Second, visual grids corresponding to each of the subtasks should be minimal modifications of the visual grids of the reference task, i.e., minimize $\Sigma_{k \in \{1,\dots,K\}} \mathcal{F}_{\text{diss}}^\mathbb{T}(\mathtt{T}^k, \mathtt{T}^{\text{ref}})$. This is based on the intuition that minimal switch in visual context is better for problem-solving (Burnett & McIntyre, 1995; Kiel, 2009); see Section 5. Third, subtasks in the progression should generally be diverse. In our implementation, we use these three properties for tie-breaking when solving Equation 2.[3]

## 3  Our Synthesis Algorithm

In this section, we present our algorithm, PROGRESSYN, for synthesizing a progression of subtasks for a reference task and its solution code $(\mathtt{T}^{\text{ref}}, \mathtt{C}^{\text{ref},*})$. PROGRESSYN builds up on two procedures: PROGRESSYN$^{\text{grids}}$ for gradually introducing visual grids as subtasks and PROGRESSYN$^{\text{single}}$ for synthesizing subtasks for a task with a single visual grid.

### 3.1  ProgresSyn$^{\text{grids}}$

Existing programming platforms such as *HackerRank* (HackerRank, 2022), *Codeforces* (Codeforces, 2022) and initiatives such as Code Hunt (Bishop et al., 2015) have test cases to guide the process of programming. In block-based visual programming, visual grids of a task are equivalent to test cases that a solution code must satisfy. We begin by describing how to construct subtasks from subsets of visual grids of a reference task and then present a procedure that generates a progression by gradually introducing the visual grids.

**Generating a progression of subtasks.** For a task $\mathtt{T}$ and a subset of visual grids $G \subseteq \mathtt{T}_{\text{vis}}$, we define a *code reduction* $\hat{\mathtt{C}}$ as the code that solves $G$ and is obtained by removing inactive branches of $\mathtt{C}^{\mathtt{T},*}$ during program execution (Ferles et al., 2017; Korel & Laski, 1988). We denote a code reduction as $\hat{\mathtt{C}} := \text{REDCODE}(G; \mathtt{T}, \mathtt{C}^{\mathtt{T},*})$. Using these elements, we define a subtask of $\mathtt{T}$ as $\hat{\mathtt{T}} := (\hat{\mathtt{T}}_\text{n} = |G|, \hat{\mathtt{T}}_{\text{vis}} = G, \hat{\mathtt{T}}_{\text{store}} = \hat{\mathtt{C}}_{\text{blocks}}, \hat{\mathtt{T}}_{\text{size}} = \hat{\mathtt{C}}_{\text{size}})$. Using these subtasks, we develop a procedure PROGRESSYN$^{\text{grids}}$ to generate our progression of subtasks. Before describing our procedure, we introduce some notation: $\Sigma^n$ denotes the collection of all permutations of the set $\{1,2,\dots,n\}$; $\sigma \in \Sigma^n$ refers to one such permutation in the collection; $\sigma_i$ refers to the $i$-th element of $\sigma$. Our procedure begins by generating a progression of subtasks for a given $\sigma \in \Sigma^{\mathtt{T}_\text{n}^{\text{ref}}}$, denoted as $\omega^\sigma$. In $\omega^\sigma$, we define

---

[3]In Equation 2, we have defined the set of progressions $\Omega$ w.r.t a single solution code of the reference task; however, $\Omega$ could be extended to account for multiple solution codes.

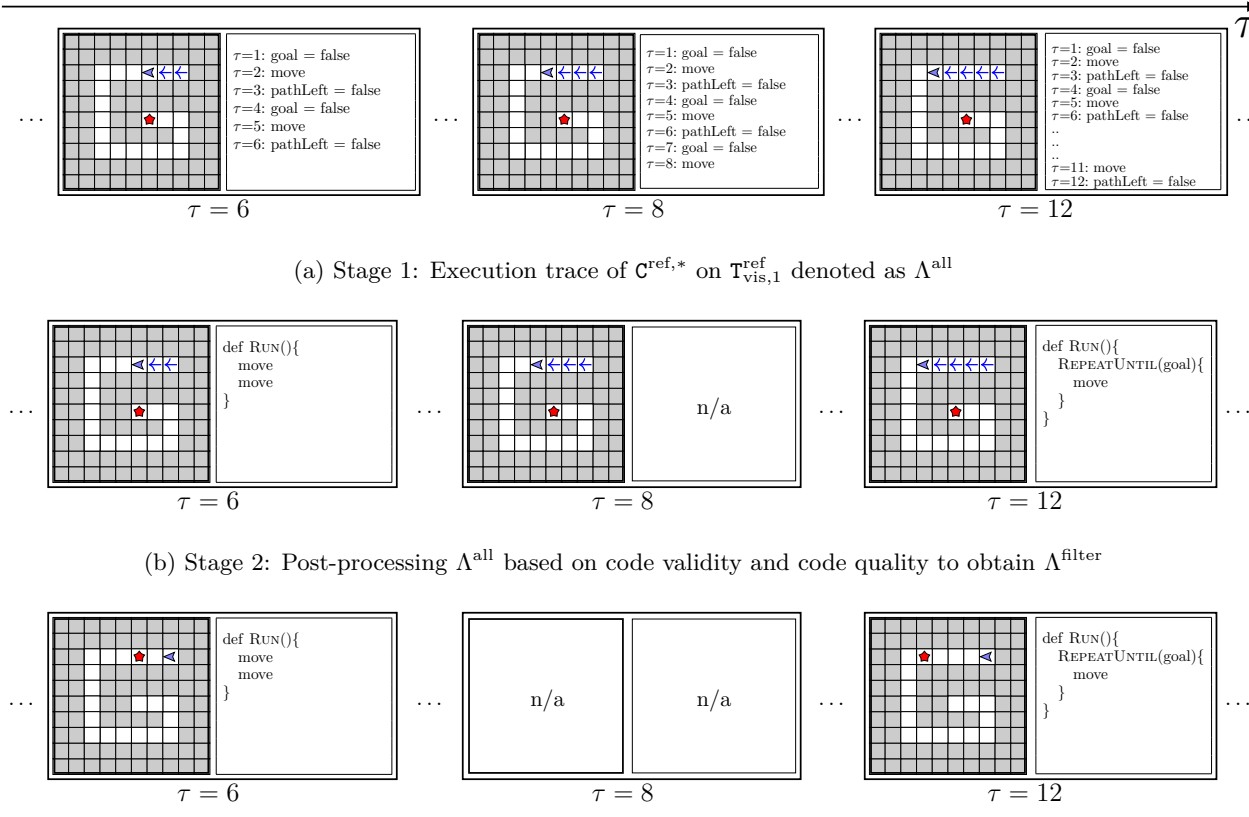

(a) Stage 1: Execution trace of $\mathtt{C}^{\mathrm{ref},*}$ on $\mathtt{T}^{\mathrm{ref}}_{\mathrm{vis},1}$ denoted as $\Lambda^{\mathrm{all}}$

(b) Stage 2: Post-processing $\Lambda^{\mathrm{all}}$ based on code validity and code quality to obtain $\Lambda^{\mathrm{filter}}$

(c) Stage 3: Modifying grids in $\Lambda^{\mathrm{filter}}$ via symbolic execution to obtain $\Lambda^{\mathrm{SE}}$

Figure 3: Illustration of three stages of ProgresSyn$^{\mathrm{single}}$ on reference task shown in Figure 1a. "n/a" denotes invalid codes and tasks. In Stage 4, we select a progression of $K' = 3$ subtasks shown in Figures 1b, 1c, and 1d. See the details in Section 3.2.

the $k$-th subtask with visual grids $G = \{\mathtt{T}^{\mathrm{ref}}_{\mathrm{vis},\sigma_i}\}_{i=1,\ldots,k}$ and solution code $\mathtt{C} = \textsc{RedCode}(G; \mathtt{T}^{\mathrm{ref}}, \mathtt{C}^{\mathrm{ref}})$ as $\mathtt{T} := (|G|, G, \mathtt{C}_{\mathrm{blocks}}, \mathtt{C}_{\mathrm{size}})$. Our procedure then generates a set of progressions of subtasks as $\Omega(\mathtt{T}^{\mathrm{ref}}, \mathtt{C}^{\mathrm{ref},*}, \mathtt{T}^{\mathrm{ref}}_{\mathrm{n}}) := \{\omega^{\sigma}\}_{\sigma \in \Sigma^{\mathtt{T}^{\mathrm{ref}}_{\mathrm{n}}}}$. Using $\Omega(\mathtt{T}^{\mathrm{ref}}, \mathtt{C}^{\mathrm{ref},*}, \mathtt{T}^{\mathrm{ref}}_{\mathrm{n}})$, we obtain our optimized progression of subtasks for $\mathtt{T}^{\mathrm{ref}}$ based on Equation 2. When $\mathtt{T}^{\mathrm{ref}}_{\mathrm{n}}$ is small, we can optimize for Equation 2 by enumerating all possible elements from $\Omega$. When $\mathtt{T}^{\mathrm{ref}}_{\mathrm{n}}$ is large, we can use greedy optimization strategies; we present further details in Appendix C.1.

**Need for more fine-grained subtasks.** However, generating a progression of subtasks for reference task $\mathtt{T}^{\mathrm{ref}}$ using only subsets of its visual grids is limiting because of the following reasons: (a) When $\mathtt{T}^{\mathrm{ref}}_{\mathrm{n}} = 1$, our procedure would return $\mathtt{T}^{\mathrm{ref}}$ as the only subtask which would not serve our intended purpose of reducing the complexity of the reference task; (b) Reference tasks with $\mathtt{T}^{\mathrm{ref}}_{\mathrm{n}} > 1$ could have all its visual grids requiring high code coverage of $\mathtt{C}^{\mathrm{ref},*}$, resulting in high complexity of all reduced codes and hence high complexity of all subtasks. To overcome these limitations, there is a need for a more fine-grained procedure to synthesize subtasks as discussed next.

## 3.2 ProgresSyn$^{\mathrm{single}}$

Next, we describe a procedure ProgresSyn$^{\mathrm{single}}$ for obtaining subtasks for a single-grid task $\mathtt{T}^{\mathrm{ref}}$ (i.e., $\mathtt{T}^{\mathrm{ref}}_{\mathrm{n}} = 1$) and its solution code $\mathtt{C}^{\mathrm{ref},*}$. Our procedure first "linearizes" $\mathtt{C}^{\mathrm{ref},*}$ by obtaining its execution trace on $\mathtt{T}^{\mathrm{ref}}_{\mathrm{vis},1}$. This makes it easier to segment and generate well-spaced codes for the progression of subtasks. Then, we process elements of the trace to generate codes which serve as solution codes of the subtasks. Next, we synthesize

visual grids from the solution codes via minimal modifications w.r.t. to $\mathtt{T}^{\mathrm{ref}}_{\mathrm{vis},1}$ using symbolic execution. Finally, we select a sequence of $K'$ subtasks to obtain our progression. These four stages are described below.

**Stage 1: Execution trace on the single grid (Figure 3a).** This stage "linearizes" the solution code $\mathtt{C}^{\mathrm{ref},*}$ by obtaining the full execution trace of this code on single visual grid $\mathtt{T}^{\mathrm{ref}}_{\mathrm{vis},1}$. We denote the execution trace as the sequence $\Lambda^{\mathrm{all}}(\mathtt{T}^{\mathrm{ref}}, \mathtt{C}^{\mathrm{ref},*}) := ((\lambda^{\mathtt{T}^{\mathrm{ref}}}_\tau, \lambda^{\mathtt{C}^{\mathrm{ref},*}}_\tau))_{\tau=1,\ldots,M^{\mathrm{all}}}$ where $M^{\mathrm{all}}$ is the total steps in the execution, $\lambda^{\mathtt{C}^{\mathrm{ref},*}}_\tau$ is the sequence of code commands executed till step $\tau$, and $\lambda^{\mathtt{T}^{\mathrm{ref}}}_\tau$ is the state of $\mathtt{T}^{\mathrm{ref}}_{\mathrm{vis},1}$ at time-step $\tau$ after $\lambda^{\mathtt{C}^{\mathrm{ref},*}}_\tau$ is executed.

**Stage 2: Post-processing the trace based on code validity/quality (Figure 3b).** This stage filters the execution trace and generates potential solution codes of the subtasks. We begin by filtering those elements of the trace whose code commands lead to invalid codes. For example, in Figure 3b, code at step $\tau = 8$ is filtered because the corresponding code commands in Stage 1 terminate on `move`, which is in the middle of the body of loop `REPEATUNTIL` of $\mathtt{C}^{\mathrm{ref},*}$. For the remaining elements, we generate codes from their code commands – these codes eventually serve as solution codes of the subtasks. This stage provides a new sequence denoted as $\Lambda^{\mathrm{filter}}(\mathtt{T}^{\mathrm{ref}}, \mathtt{C}^{\mathrm{ref},*}) := ((\lambda^{\mathtt{T}^{\mathrm{ref}}}_\tau, \mathtt{C}^\tau))_{m=1,\ldots,M^{\mathrm{filter}}}$ where $M^{\mathrm{filter}} \leq M^{\mathrm{all}}$. Further details of this stage can be found in Appendix C.2.

**Stage 3: Modifying grids in the trace via symbolic execution (Figure 3c).** In this stage, we generate task grids for each of the codes from the sequence obtained in Stage 2. We achieve this using symbolic execution techniques (King, 1976; Ahmed et al., 2020). Specifically, during symbolic execution, we make minimal modifications to the grids of the subtasks w.r.t. $\mathtt{T}^{\mathrm{ref}}_{\mathrm{vis},1}$ and generate high quality subtasks. For example, consider step $\tau = 6$ in Figures 3b and 3c . After symbolic execution on the code from Figure 3b, we obtain the grid in Figure 3c. Observe how the "goal" (red star) is moved to the final location of "avatar" on the grid from Figure 3b to generate the grid in Figure 3c. This stage provides a new sequence of subtasks denoted as $\Lambda^{\mathrm{SE}}(\mathtt{T}^{\mathrm{ref}}, \mathtt{C}^{\mathrm{ref},*}) := ((\mathtt{T}^\tau, \mathtt{C}^\tau))_{\tau=1,\ldots,M^{\mathrm{SE}}}$ where $M^{\mathrm{SE}} \leq M^{\mathrm{filter}}$. Further details of this stage can be found in Appendix C.2.

**Stage 4: Generating subtasks via subsequence selection.** In the final stage, we generate a progression of $K'$ subtasks. From $\Lambda^{\mathrm{SE}}$, we obtain a set of all subsequences of length $K'$ denoted as $\Omega(\mathtt{T}^{\mathrm{ref}}, \mathtt{C}^{\mathrm{ref},*}, K')$. Using $\Omega$, we optimize for Equation 2 to obtain our final progression. Further details of this stage can be found in Appendix C.2.

### 3.3 ProgresSyn

Our synthesis algorithm PROGRESSYN is a combination of procedures described in Sections 3.1 and 3.2. Algorithm 1 provides an overview of PROGRESSYN and we present the implementation details in Appendix C.

---
**Algorithm 1** PROGRESSYN
---
1: **function** PROGRESSYN($\mathtt{T}^{\mathrm{ref}}, \mathtt{C}^{\mathrm{ref},*}, K$)
2:      $\Omega \leftarrow \{\}$; $\Sigma^{\mathtt{T}^{\mathrm{ref}}_{\mathrm{n}}} \leftarrow$ permutations of $\{1, 2, \ldots, \mathtt{T}^{\mathrm{ref}}_{\mathrm{n}}\}$
3:      **for** $\sigma \in \Sigma^{\mathtt{T}^{\mathrm{ref}}_{\mathrm{n}}}$ **do**
4:          $\mathtt{C}^{\mathrm{single},*} \leftarrow$ REDCODE($\{\mathtt{T}^{\mathrm{ref}}_{\mathrm{vis},\sigma_1}\}; \mathtt{T}^{\mathrm{ref}}, \mathtt{C}^{\mathrm{ref},*}$)
5:          $\mathtt{T}^{\mathrm{single}} := (1, \{\mathtt{T}^{\mathrm{ref}}_{\mathrm{vis},\sigma_1}\}, \mathtt{C}^{\mathrm{single},*}_{\mathrm{blocks}}, \mathtt{C}^{\mathrm{single},*}_{\mathrm{size}})$
6:          $\omega 1 \leftarrow$ PROGRESSYN$^{\mathrm{single}}$($\mathtt{T}^{\mathrm{single}}, \mathtt{C}^{\mathrm{single},*}, K'$) where $K' = K - \mathtt{T}^{\mathrm{ref}}_{\mathrm{n}} + 1$
7:          $\omega 2 \leftarrow$ PROGRESSYN$^{\mathrm{grids}}$($\mathtt{T}^{\mathrm{ref}}, \mathtt{C}^{\mathrm{ref},*}, \sigma$), i.e., progression for a given permutation $\sigma$
8:          $\omega \leftarrow$ Concatenate $\omega 1$ with $\omega 2$ (after removing the common element); add $\omega$ to $\Omega$
9:      **Return** $\omega^* \in \Omega$ **by optimizing for Equation 2**
---

## 4 Experimental Validation and Evaluation

In this section, we validate the quality of subtasks generated by PROGRESSYN according to the metrics introduced in Section 2.2 and evaluate their utility in assisting AI agents towards solving block-based programming tasks. We evaluate the subtasks in the context of AI agents as a warm up before evaluating

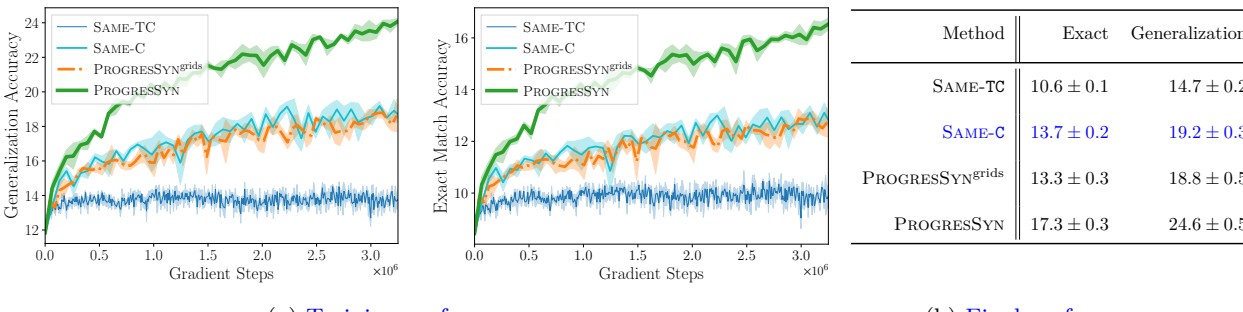

(a) Training performance

(b) Final performance

Figure 4: This figure shows the performance of a neural program synthesizer based on Bunel et al. (2018) and trained by augmenting their dataset with subtasks generated using different methods. Results are reported as mean and standard deviation over 3 random seeds. Plots in **(a)** show the generalization and exact match accuracy during training when evaluated on the validation dataset of 2500 tasks from Bunel et al. (2018); the x-axis shows the number of gradient steps. Table in **(b)** shows the final performance of the model obtained for each method when evaluated on the test dataset of 2500 tasks from Bunel et al. (2018).

them with novice programmers (discussed subsequently in Section 5). We consider two different settings: (i) validating the diversity of subtasks in improving the efficacy of neural program synthesizers; (ii) analyzing the utility of progression of subtasks w.r.t. code complexity in guiding search-based agents to solve a specific task.

### 4.1 Neural Program Synthesis

A neural program synthesizer, at a high level, is a neural network based model that takes a visual task as input, and sequentially synthesizes its solution code (Gulwani et al., 2017; Chen et al., 2019). We model our synthesizer using reinforcement learning methods based on the work of Bunel et al. (2018). However, it is challenging to train such synthesizers due to the sparsity of learning signals. Inspired by the work of Eysenbach et al. (2021), we augment the training dataset using our synthesized subtasks to increase its diversity, thereby improving the strength of the learning signals. This also aligns with our objective of evaluating the helpfulness of the subtasks in learning to solve a given reference task.

**Setup.** Our experimental setup is based on the work of Bunel et al. (2018). Specifically, we borrow their training dataset of 10000 *Karel programming tasks*, and generate variants of it using different subtasking methods. The training process uses their reinforcement learning (RL)-based training component, starting from the same pretrained model for all methods obtained using the supervised learning component. As our goal was to evaluate the helpfulness of subtasks in learning to solve a given reference task, the RL-based training component is more suitable for our evaluation (as it implicitly helps in dealing with exploration). We measure the gradient steps during training as the agent learns to synthesize solution codes. While we borrow the experimental setup from the work of Bunel et al. (2018), our setup differs from them in two major ways, as discussed next. Firstly, our approach considers good quality tasks where the solution code has full coverage, and we pruned tasks from the training dataset of 10000 tasks where the solution code did not have full coverage w.r.t. the task (e.g., tasks whose solution code had redundant blocks). Hence, the final training dataset comprised of 7300 Karel tasks. Secondly, the RL-based training component, central to our evaluation, is much slower than the supervised training component. Specifically, our training took 8 days for 3.25 million gradient steps. To speed up the training process, we did not use beam search for RL-training. We present additional details of our setup and dataset in Appendix D.1.

**Methods evaluated.** We generate variants of the training dataset by augmenting it with subtasks synthesized for each training task using the following methods: PROGRESSYN, PROGRESSYN$^{\text{grids}}$, SAME-TC, and SAME-C. Methods PROGRESSYN and PROGRESSYN$^{\text{grids}}$ are described in Section 3; in our setup, each training task $T^{\text{ref}}$ has $T_n^{\text{ref}} = 6$ and we use $K' = 4$ with $K = 9$. SAME-TC simply contains $K$ copies of each training task. SAME-C is a variant of PROGRESSYN that uses a different version of PROGRESSYN$^{\text{single}}$ component. At a high level, SAME-C modifies PROGRESSYN$^{\text{single}}$ to synthesize subtasks from the execution

| Method | Fraction succeeded | | | | | |
|---|---|---|---|---|---|---|
| | All | $H_{08}$ | $H_{16}$ | $H_{12}$ | $K_{Diag}$ | $K_{sgStair}$ |
| DEFAULT | 0.018 | 0.004 | 0.000 | 0.084 | 0.000 | 0.000 |
| SAME | 0.055 | 0.020 | 0.004 | 0.238 | 0.012 | 0.000 |
| CRAFTED | 0.783 | 0.996 | 0.426 | 0.964 | 0.946 | 0.628 |
| PROGRESSYN | 0.836 | 0.842 | 0.722 | 0.998 | 0.992 | 0.628 |

Figure 5: The table shows the results of the study with Monte-Carlo agents on reference tasks $H_{08}$, $H_{12}$, $H_{16}$, $K_{Diag}$, and $K_{sgStair}$. We report the average success rate over 500 agents (each generated from a different random seed) on these reference tasks under the column "Fraction succeeded" (higher scores being better)

trace of a given single-grid task s.t. their solution codes remain the same as that of the task; moreover, it ensures diversity of the visual grids of the subtasks by selecting points uniformly along the execution trace.

**Validation and evaluation.** To validate the diversity of the training dataset augmented with subtasks generated by different methods, we report the number of unique code complexity values encountered: 104 for PROGRESSYN, 89 for PROGRESSYN$^{grids}$, 78 for SAME-TC, and 89 for SAME-C. Next, we report the results of training synthesizers using augmented datasets in Figure 4, averaged over 3 random seeds. The plots show that agents trained with a more diverse training dataset have a steeper learning curve. The final performance of agents trained with PROGRESSYN is over 60% higher than those trained with SAME-TC. The final performance of PROGRESSYN is also higher than that of PROGRESSYN$^{grids}$ highlighting the utility of more fine-grained subtasking. Furthermore, PROGRESSYN outperforms SAME-C, indicating the importance of well-spaced code complexity in the progression.

## 4.2 Search-based Agents

**Setup.** In this experiment, we aim to showcase the utility of our progression of subtasks w.r.t. code complexity in guiding agents to solve a specific task. In particular, we consider Monte-Carlo agents designed to solve a reference task $T^{ref}$. We evaluate the performance on five reference tasks based on real-world tasks from *Hour of Code: Maze Challenge* by *Code.org* (Code.org, 2022c) and *CodeHS.com* (CodeHS, 2022). Specifically, these tasks are the following: Maze08 ($\mathcal{F}_{complex}^{\mathbb{T}} = 1005$), Maze12 ($\mathcal{F}_{complex}^{\mathbb{T}} = 1005$), Maze16 ($\mathcal{F}_{complex}^{\mathbb{T}} = 2004$), DIAGONAL ($\mathcal{F}_{complex}^{\mathbb{T}} = 1006$), and single-grid STAIRWAY ($\mathcal{F}_{complex}^{\mathbb{T}} = 2007$). Henceforth, we refer to these tasks as $H_{08}$, $H_{12}$, $H_{16}$, $K_{Diag}$, and $K_{sgStair}$ respectively. $H_{16}$ is illustrated in Figure 1a. The design of the setup as described below is to simulate a participation session in our user study (see Section 5). In our setup, the goal of the agent is to navigate the space of codes to reach the solution code $C^{ref,*}$ via successive code edits and starting from empty code {RUN}. We guide the agent through the code space via a progression of $K$ subtasks for $T^{ref}$. Specifically, we divide the agent's search process into $K$ stages, corresponding to $K$ subtasks of the progression. Each stage is modeled as a deterministic finite-horizon Markov Decision Process (MDP) where states represent different partial codes and actions are possible edits in the code space (such as adding/deleting/updating a code block).[4] The agent follows an $\epsilon$-greedy policy and uses Monte-Carlo updates for its value function after each episode (Sutton & Barto, 2018). Each stage lasts for $N$ episodes. Further details about the agent are in Appendix E.

**Methods evaluated.** We evaluate the performance of PROGRESSYN in comparison to three baseline methods: DEFAULT, SAME, CRAFTED. For all five reference tasks, $T_n^{ref} = 1$; we use $K' = 3$ and $K = 3$. Next, we describe the baseline methods; importantly, we will further use these methods for the user study in Section 5. Method DEFAULT is a default setting where the agent directly solves the reference task with no

---

[4]Our search-based agents do not encode the visual grids of the tasks as part of the state space because this would require new neural architectures beyond the scope of this work. One could have also used the neural architectures from the neural program synthesizers as a basis for these agents. However, these architectures build the code sequentially, whereas our search-based agents need to make non-trivial edits (such as nesting existing blocks within a while block) in their codes to leverage the benefits of subtasks in a progression.

| Method | Total participants | | | Fraction succeeded | | |
|---|---|---|---|---|---|---|
| | Both | $H_{08}$ | $H_{16}$ | Both | $H_{08}$ | $H_{16}$ |
| DEFAULT | 114 | 57 | 57 | 0.605 | 0.807 | 0.403 |
| SAME | 230 | 116 | 114 | 0.669 | 0.845 | 0.491 |
| SAME-TC | 114 | 57 | 57 | 0.667 | 0.842 | 0.491 |
| SAME-C | 116 | 59 | 57 | 0.672 | 0.847 | 0.491 |
| CRAFTED | 235 | 117 | 118 | 0.647 | 0.838 | 0.458 |
| PROGRESSYN | 117 | 58 | 59 | 0.701 | 0.862 | 0.542 |

Figure 6: The table shows results of the study with novice programmers on tasks $H_{08}$ and $H_{16}$. We report the success rate on different reference tasks under the column "Fraction succeeded" (higher scores being better).

subtasks. Methods SAME-TC and SAME-C (collectively called SAME) generate a progression of 3 subtasks which are not well-spaced w.r.t. their code complexity. Specifically, SAME-TC contains 3 subtasks all of which are same as the reference task. SAME-C alters only the visual grids of subtasks in the progression while $\forall k \in \{1, 2, 3\}, C^{k,*} = C^{\text{ref},*}$. Methods CRAFTED-v1 and CRAFTED-v2 (collectively called CRAFTED) generate a handcrafted progression of 3 subtasks, that are well spaced w.r.t. their code complexity but have task grids that do not retain the visual context of the reference task. Figure 7 illustrate subtasks generated by these methods for reference tasks $H_{16}$. We note that method DEFAULT only comprises a single stage as it does not have any subtasks; all other methods comprise 3 stages corresponding to $K = 3$ subtasks in their progression.

**Validation and evaluation.** We validate the progression of subtasks generated by different methods on these two reference tasks based on the maximum code complexity jump in their progression. As reported in Appendix E.3, we find that progressions synthesized by PROGRESSYN have the lowest maximum code complexity jump; detailed analysis across different metrics is provided in the appendix. Next, we report the results about the fraction of Monte-Carlo agents that succeeded in solving reference tasks, guided by different subtasking methods; see Figure 5. We find that PROGRESSYN and CRAFTED perform significantly better than SAME, which indicates the utility of using progression of subtasks with well-spaced code complexity to guide problem-solving. We note that the Monte-Carlo agents only operate on the code space ignoring the visual grids of the tasks (see Footnote 4). Further results and analysis are presented in Appendix E.4.

## 5 Assisting Novice Human Programmers

**Setup.** Finally, we evaluate the effectiveness of our synthesis algorithm, PROGRESSYN, in assisting a novice programmer to solve a specific reference task. We recruited participants for the study from Amazon Mechanical Turk; an IRB approval had been obtained for the study. The participants were US-based adults, without expertise in block-based visual programming. Due to the costs involved (over 4 USD per task for a participant), we used two block-based programming tasks for the study: $\{H_{08}, H_{16}\}$.

We developed a web app for the study. The app uses the publicly available toolkit of Blockly Games (Games, 2022) and provides an interface for a participant to solve a block-based visual programming task through a progression of subtasks. Before logging into the app, each participant was encouraged to watch a 4 minute instructional video about block-based programming to familiarize themselves with the platform. After logging into the app, a participant is assigned a reference task $T^{\text{ref}} \in \{H_{08}, H_{16}\}$ and one of the subtasking methods at random. These elements constituted a "session" for a participant. Specifically, a participation session comprised of the following steps: (i) Step 1: The participant is shown the reference task, and given 10 attempts to solve it. If they are successful, they exit the platform; otherwise, they proceed to the next step; (ii) Steps 2a / 2b: The participant is presented with the first $K - 1$ (i.e., 2) subtasks from the progression synthesized by the assigned subtasking method, and given 10 attempts to solve each subtask; (iii) Step 3:

The participant is presented with the reference task from Step 1 again, and given 10 attempts to solve it. Each of these steps are illustrated in Appendix F.1.

**Methods evaluated.** Similar to our evaluation with Monte-Carlo agents (described in Section 4), we evaluate the performance of ProgresSyn in comparison to the following methods: Default, Same, Crafted. For both our reference tasks, $\mathtt{T}_n^{ref} = 1$; we use $K' = 3$ and $K = 3$. Method Default is the setting where the participant is presented with $\mathtt{T}^{ref}$ and given 10 tries to solve it (only Step 1 of the participation session). Methods Same-TC and Same-C (collectively called Same) generate a progression of 3 subtasks which are not well spaced w.r.t their code complexity. Specifically, Same-TC contains 3 subtasks all of which are the same as the reference task. Same-C minimally alters the visual grids of the subtasks while their solution codes remains the same as that of the reference task. Methods Crafted-v1 and Crafted-v2 (collectively called Crafted) generate a handcrafted progression of 3 subtasks, that are well spaced w.r.t. their code complexity but have task grids that do not retain the visual context of the reference task. We note that a participant spends up to 40 problem-solving attempts in all the subtasking methods (Same, Crafted, ProgresSyn) and up to 10 problem-solving attempts without subtasking (Default). Further details can be found in Appendix F. Next, we present an overview and key results of our study.

**Research questions.** We center our user study around the following research questions *(RQs)* to measure the efficacy of ProgresSyn: (i) *RQ1: Usefulness of subtasking.* Does solving a progression of subtasks increase success rate on the reference task? (ii) *RQ2: Well-spaced code complexity.* Do progressions with subtasks that are well spaced w.r.t. their code complexity improve the success rate more in comparison to progressions that violate this property? (iii) *RQ3: Retaining visual context of the reference task.* Do progression of subtasks that retain visual context of the reference task in their grids improve the success rate more in comparison to progressions which violate this property?

**Results.** We present detailed results in Figure 6. In total, we had over 500 participation sessions across two tasks. To validate the *usefulness of subtasking (RQ1)*, we compare the success rate on the reference task for methods Same-TC and ProgresSyn. We find a 5% increase in the success rate for ProgresSyn, suggesting the usefulness of subtasks in problem-solving. To investigate the effect of *well-spaced code complexity (RQ2)* in a progression, we compare the success rates for Same-C and ProgresSyn. We find that ProgresSyn outperforms Same-C by 4.5%. This suggests the importance of using progressions with well-spaced code complexity. To investigate the effect of *retaining visual context of the reference task (RQ3)* in the progression, we compare Crafted and ProgresSyn. We find a 8% increase in success rate for ProgresSyn compared to Crafted, suggesting the importance of retaining the visual context. Furthermore, we find that Same also outperforms Crafted. We hypothesize that this is because Crafted synthesizes subtasks that are visually very different from the reference task, possibly making the progression more confusing and leading to lower success on the reference task. We provide a more detailed analysis in Appendix F.3.

# 6 Concluding Discussion

Here, we discuss a few limitations of our current work and outline a plan for future work to address them. For neural program synthesizers in Section 4, our training process is time consuming as it uses datasets augmented with the synthesized subtasks. It would be interesting to see how our synthesized subtasks can be used to explicitly design intermediate rewards to obtain similar performance gains with lower training time. Our current user study in Section 5 is limited by the fact that it is conducted with adult novice programmers. In the future, it would be important to conduct longitudinal studies with real students using our methodology and measure its pedagogical value (Margulieux et al., 2020).

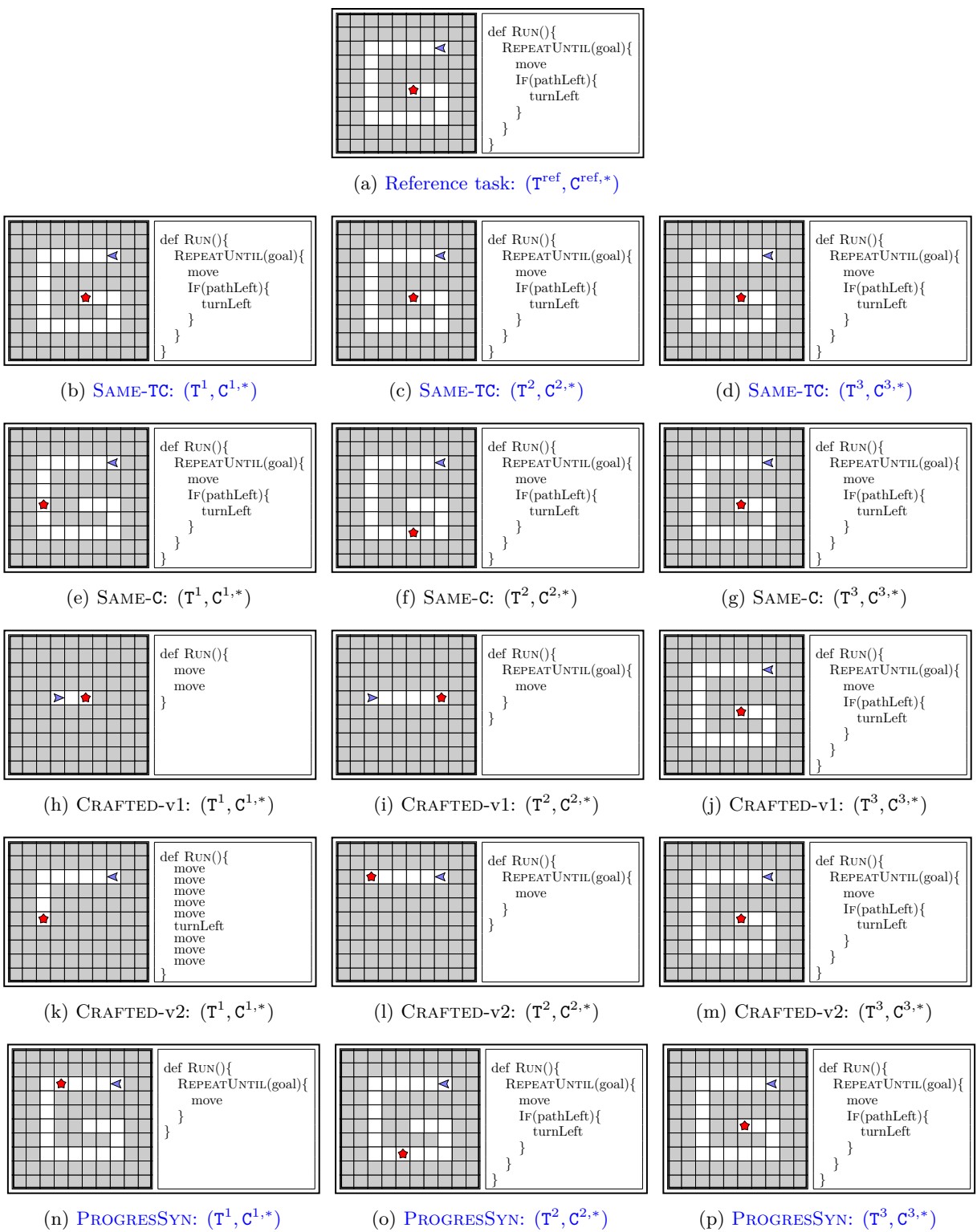

Figure 7: Illustration of subtasks synthesized by baseline methods SAME-TC, SAME-C, CRAFTED-v1, CRAFTED-v2, and PROGRESSYN for the reference task $H_{16}$ shown in (a). Each method synthesizes a progression of three subtasks. SAME-TC and SAME-C are collectively called SAME; CRAFTED-v1 and CRAFTED-v2 are collectively called CRAFTED.

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

## A  List of Appendices

In this section, we provide a brief description of the content in the appendices of the paper.

- Appendix B provides a table of notations used throughout the paper.
- Appendix C provides additional details about our synthesis algorithm PROGRESSYN. (Section 3)
- Appendix D provides additional details and results on improving the efficacy of neural program synthesizers using our subtasking algorithm. (Section 4.1)
- Appendix E provides additional details and results on improving the efficacy of search-based agents using our subtasking algorithm. (Section 4.2)
- Appendix F provides additional details about our study with novice human programmers. (Section 5)

## B  Table of Notations (Section 2)

See Figure 6 for the complete list of notations.

| Notation | Description |
|---:|:---|
| $\mathtt{T}$ | Task identifier |
| $\mathtt{T_n}$ | Number of visual grids in task $\mathtt{T}$ |
| $\mathtt{T_{vis}}$ | The set of visual grids of task $\mathtt{T}$ |
| $\mathtt{T_{store}}$ | The types of code blocks available to solve the task $\mathtt{T}$ |
| $\mathtt{T_{size}}$ | The maximum number of code blocks allowed in the solution code of task $\mathtt{T}$ |
| $\mathbb{T}$ | Task space |
| $\mathcal{F}^{\mathbb{T}}_{complex} : \mathbb{T} \to \mathbb{R}$ | Function to measure the complexity of a task |
| $\mathcal{F}^{\mathbb{T}}_{diss} : \mathbb{T} \times \mathbb{T} \to \mathbb{R}$ | Function to measure the dissimilarity between two tasks |
| $\mathtt{C}$ | Code indentifier |
| $\mathtt{C_{depth}}$ | Depth of the Abstract Syntax Tree (AST) of a code $\mathtt{C}$ |
| $\mathtt{C_{size}}$ | Number of code blocks in code $\mathtt{C}$ |
| $\mathtt{C_{blocks}}$ | Types of code blocks in code $\mathtt{C}$ |
| $\mathbb{C}$ | Code space |
| $\mathcal{F}^{\mathbb{C}}_{complex} : \mathbb{C} \to \mathbb{R}$ | Function to measure the complexity of a code |
| $\mathbb{C}_{\mathtt{T}}$ | Set of all solution codes of task $\mathtt{T}$ |
| $K$ | Fixed budget on the number of subtasks for a reference task |
| $\omega$ | A specific progression of subtasks for a reference task |
| $\Omega$ | Set of all progressions of subtasks for a reference task |
| $\mathcal{F}^{\Omega}_{complex}$ | Function to measure the complexity of a progression of subtasks for a reference task |
| $\lambda^{\mathtt{T}}_{\tau}$ | State of the first visual grid of task $\mathtt{T}$ at time-step $\tau$ |
| $\lambda^{\mathtt{C}}_{\tau}$ | Partial code obtained from code $\mathtt{C}$, after it is executed till time-step $\tau$ |
| $\Lambda^{\mathrm{all}}$ | Execution trace of a code on a single grid of a task; obtained after Stage 1 of PROGRESSYN$^{\mathrm{single}}$ |
| $\Lambda^{\mathrm{filter}}$ | Filtered execution trace of a code on a single grid of a task; obtained after Stage 2 of PROGRESSYN$^{\mathrm{single}}$ |
| $\Lambda^{\mathrm{SE}}$ | Modified execution trace of a code on a single grid of a task; obtained after Stage 3 of PROGRESSYN$^{\mathrm{single}}$ |
| $\Sigma^{\mathrm{p}}$ | set of all permutations of $\{1, 2, \ldots, p\}$ |
| $\sigma$ | A specific permutation from the set of permutations $\Sigma^{\mathrm{p}}$ |

Figure 6:  Table of notations

# C    Our Synthesis Algorithm (Section 3)

In this section, we discuss additional details of our algorithm PROGRESSYN. We begin by discussing additional details of the procedure PROGRESSYN$^{\text{grids}}$. After that, we discuss additional details of our algorithm to synthesize a progression of subtasks for a reference task with a single visual grid (PROGRESSYN$^{\text{single}}$). Finally, we present the detailed algorithm PROGRESSYN and share details of our implementation with pointers to specific code files.

## C.1    ProgresSyn$^{\text{grids}}$: Additional Details (Section 3.1)

Next, we present additional details of PROGRESSYN$^{\text{grids}}$, discussed in Section 3.1 . Specifically, we discuss the optimization strategies one can adopt for solving Equation 2 when the number of visual grids $\mathtt{T}_{\text{n}}^{\text{ref}}$, of a task $\mathtt{T}^{\text{ref}}$ is large. In this case, we can decide the sequence of subtasks generated in the final progression using a greedy strategy. We can use the degree of code coverage to generate the sequence. We can begin with grids corresponding to maximum code coverage of the solution code, and sequentially add the remaining grids..

When $\mathtt{T}_{\text{n}}^{\text{ref}}$ is small (e.g., $\leq 6$ in domains we are considering – such as the task shown in Figure 2a in Section 1), one can optimize for Equation 2 by enumerating all possible elements of $\Omega$.

## C.2    ProgresSyn$^{\text{single}}$: Additional Details (Section 3.2)

Next, we present additional details for each of the four stages of PROGRESSYN$^{\text{single}}$, for a reference task $\mathtt{T}^{\text{ref}}$ and its solution code $\mathtt{C}^{\text{ref},*}$, where $\mathtt{T}_{\text{n}}^{\text{ref}} = 1$.

**Stage 1: Execution trace of code on the single visual grid (Figure 3a).** In this stage, we obtain the full execution trace $\Lambda^{\text{all}}(\mathtt{T}^{\text{ref}}, \mathtt{C}^{\text{ref},*})$ of the solution code $\mathtt{C}^{\text{ref},*}$ on the visual grid $\mathtt{T}_{\text{vis},1}^{\text{ref}}$. We obtain the trace by instrumenting a Karel interpreter (which is used for executing Karel programs) as it executes $\mathtt{C}^{\text{ref},*}$.

**Stage 2: Post-processing the trace based on code validity and code quality (Figure 3b 3b).** In this stage we post-process the execution trace obtained in Stage 1, based on code validity and code quality, to obtain $\Lambda^{\text{filter}}(\mathtt{T}^{\text{ref}}, \mathtt{C}^{\text{ref},*})$. Specifically, we filter invalid codes from the trace whose code command sequence does the following: The code command sequence terminates on a command/code block that occurs in the middle of a loop/conditional body of $\mathtt{C}^{\text{ref},*}$. See Figure 3b, where the code at step $\tau = 8$ is filtered as its corresponding sequence of code commands in Figure 3a terminates on code block move, which is in the middle of the body of the REPEATUNTIL construct of the solution code $\mathtt{C}^{\text{ref},*}$ (shown in Figure 1a). A loop refers to the code constructs, REPEATUNTIL, WHILE and REPEAT. A conditional refers to the code constructs, IF and IFELSE.

From the remaining code commands we generate concrete codes. To generate the concrete codes, we use compiler information and the AST structure of the code. We instrument the compiler to maintain information on which branches of the AST are executed in the current sequence of code commands. Using this information, we can directly convert the sequence of commands into code. However, to maintain code quality, we insert a loop construct in the code only when the body of the loop is executed more than once in the code command sequence; when there are conditional branches inside the loop body, we add the loop construct when at least one of the branches is executed more than once. For the other codes, we retain the unrolled loop body, without the loop construct.

**Stage 3: Modifying grids in the trace via symbolic execution (Figure 3c).** In this stage, we synthesize task grids for each of the codes from the sequence obtained in Stage 2. We denote the final sequence after this stage as $\Lambda^{\text{SE}}(\mathtt{T}^{\text{ref}}, \mathtt{C}^{\text{ref},*})$. We obtain high quality visual task grids for each of the codes using symbolic execution techniques and constraint solvers. Specifically, during symbolic execution we make minimal modifications to the visual grids of the subtasks w.r.t. the reference task $\mathtt{T}^{\text{ref}}$, to generate valid tasks. Next, we describe the key ideas behind this stage.

In the sequence $\Lambda^{\text{filter}}$ obtained from Stage 2, some of the visual grid and code pairs are inconsistent (i.e., the code is not a valid solution for the visual grid) as the code execution is terminated prematurely at those steps. For example, in Figure 3, step $\tau = 12$ shows a scenario where the code execution has terminated while the avatar (purple dart) has not reached the goal (red star) yet on the visual grid. To resolve this inconsistency,

we use the trace on the visual grid as a specific path in the symbolic execution of the corresponding code. The trace determines the boolean values of the conditionals (like If, IfElse, While) of the code during symbolic execution. Combined with the trajectory of the avatar's locations, this trace gives us a set of constraints over a subset of grid cells of the visual task grid. If an assignment satisfying all the constraints does not exist, we invalidate the code and it is eliminated from the sequence along with the visual grid. Otherwise, we use the assignment values for the constrained grid cells. For the grid cells without any constraints, we retain their values from the visual grid of the reference task to ensure minimal modification (see Figure 3c). Limiting the computation to the execution trace and the avatar's trajectory on the visual grid not only allows us to retain the visual context of the reference task, but also makes this operation computationally less expensive.

**Stage 4: Generating subtasks via subsequence selection.** In this final stage, we obtain a progression of $K'$ subtasks, from the set of all subsequences of length $K'$ from $\Lambda^{\text{SE}}$. We denote this set of sequences as $\Omega(\mathtt{T}^{\text{ref}}, \mathtt{C}^{\text{ref},*}, K')$. Using $\Omega$, we optimize for Equation 2 to obtain our final progression. Specifically, we apply techniques of dynamic programming to optimize for Equation 2 and select a sequence of $K'$ elements from $\Lambda^{\text{SE}}$. This way, we do not need to enumerate over all elements of $\Omega$. While applying dynamic programming, we ensure that the following desirable properties are part of our synthesized progression of tasks: (i) maximize $\Sigma_{k \in \{1,\dots,K'\}} \mathcal{F}_{\text{qual}}^{\mathbb{T}}(\mathtt{T}^k)$; (ii) minimize $\Sigma_{k \in \{1,\dots,K'\}} \mathcal{F}_{\text{diss}}^{\mathbb{T}}(\mathtt{T}^k, \mathtt{T}^{\text{ref}})$; (iii) subtasks in the progression are diverse from each other.

## C.3 ProgresSyn: Detailed Algorithm (Section 3)

The detailed algorithm to synthesize a progression of subtasks for a given reference task $\mathtt{T}^{\text{ref}}$ and its solution code $\mathtt{C}^{\text{ref},*}$ is presented in Algorithm 2. Note that, we present only the pseudo-code of our algorithm. Our implementation optimizes the algorithm further to avoid redundant computations and run-time overheads.

**A few important points about our algorithm:**

- In our implementation of the code complexity function $\mathcal{F}_{\text{complex}}^{\mathbb{C}} = \kappa * \mathtt{C}_{\text{depth}} + \mathtt{C}_{\text{size}}$ (see Section 2.1), we set $\kappa = 1000$.
- The time-complexity of our dynamic programming routine is $O(K(M^{SE})^3)$, indicating that our algorithm is linear w.r.t. the number of subtasks $K$. Here, $M^{SE}$ is bounded by the length of the execution trace of the solution code $\mathtt{C}^*$ on task $\mathtt{T}$ (See Stage 1 of ProgresSyn$^{\text{single}}$ in Section 3.2).
- In our procedure, the number of subtasks $K$ provides a trade-off between the number of subtasks that a student must solve, and the complexity jumps between consecutive subtasks. In practice, one can assume that a teacher provides an upper and lower bound on $K$, as well as a bound on the maximum complexity jump in the sequence of subtasks. Our algorithm could be extended to find the minimal $K$ satisfying these constraints. For the reference tasks shown in Figure 1a and Figure 2a (in Section 1), we set $K' = 3$ for our single grid decomposition procedure ProgresSyn$^{\text{single}}$. For the multi-grid Karel task, Stairway (shown in Figure 2a), which had $\mathtt{T}_n^{\text{ref}} = 3$ visual grids, this resulted in $K' + \mathtt{T}_n^{\text{ref}} - 1 = 3 + 3 - 1 = 5$ subtasks.

## C.4 ProgresSyn: Implementation files

We present three demo scripts in our supplementary code files, which can be executed to obtain progression of subtasks synthesized by ProgresSyn for reference tasks $\text{H}_{08}$ (see Figure 12a), $\text{H}_{16}$ (see Figure 1a) and Stairway task from *CodeHS.com* based on the *Karel programming environment* (see Figure 2a). The scripts are located in the folder "code/algorithm_progresssyn_demo" and are as follows:

- Demo for $\text{H}_{08}$: `progressyn_hoc08.py`
- Demo for $\text{H}_{16}$: `progressyn_hoc16.py`
- Demo for Karel Stairway: `progressyn_stairway.py`

Specifically, in our supplementary code folder "code/algorithm_progressyn/subtasking/", the following functions correspond to our synthesis algorithms:

- ProgresSyn is implemented in the function `progressyn()`

---

**Algorithm 2** PROGRESSYN

---

1: **function** PROGRESSYN($\mathtt{T}^{\mathrm{ref}}, \mathtt{C}^{\mathrm{ref},*}, K$)
2:     **Initialize:**  $\Omega \leftarrow \{\}$; $\Sigma^{\mathtt{T}^{\mathrm{ref}}_{\mathrm{n}}} \leftarrow$ set of all permutations of $\{1, 2, \ldots, \mathtt{T}^{\mathrm{ref}}_{\mathrm{n}}\}$
3:     **for** $\sigma \in \Sigma^{\mathtt{T}^{\mathrm{ref}}_{\mathrm{n}}}$ **do**
4:         $\mathtt{C}^{\mathrm{single},*} \leftarrow \mathrm{REDCODE}(\{\mathtt{T}^{\mathrm{ref}}_{\mathrm{vis},\sigma_1}\}; \mathtt{T}^{\mathrm{ref}}, \mathtt{C}^{\mathrm{ref},*})$
5:         $\mathtt{T}^{\mathrm{single}} := (1, \{\mathtt{T}^{\mathrm{ref}}_{\mathrm{vis},\sigma_1}\}, \mathtt{C}^{\mathrm{single},*}_{\mathrm{blocks}}, \mathtt{C}^{\mathrm{single},*}_{\mathrm{size}})$
6:         $\omega 1 \leftarrow \mathrm{PROGRESSYN}^{\mathrm{single}}(\mathtt{T}^{\mathrm{single}}, \mathtt{C}^{\mathrm{single},*}, K')$ where $K' = K - \mathtt{T}^{\mathrm{ref}}_{\mathrm{n}} + 1$
7:         $\omega 2 \leftarrow \mathrm{PROGRESSYN}^{\mathrm{grids}}(\mathtt{T}^{\mathrm{ref}}, \mathtt{C}^{\mathrm{ref},*}, \sigma)$, i.e., progression for a given permutation $\sigma$
8:         $\omega \leftarrow$ Concatenate $\omega 1$ with $\omega 2$ (after removing the common element)
9:         Add $\omega$ to $\Omega$
10:     $\omega^* = \arg\min_{\omega \in \Omega} \mathcal{F}^{\Omega}_{\mathrm{complex}}(\omega; \mathtt{T}^{\mathrm{ref}}, \mathtt{C}^{\mathrm{ref},*}, K)$ as per Equation 2
11:     **Return** $\omega^* \in \Omega$

12: **function** PROGRESSYN$^{\mathrm{SINGLE}}$($\mathtt{T}^{\mathrm{single}}, \mathtt{C}^{\mathrm{single},*}, K'$)
13:     [Stage 1] Obtain the execution trace $\Lambda^{\mathrm{all}}(\mathtt{T}^{\mathrm{single}}, \mathtt{C}^{\mathrm{single},*})$
14:     [Stage 2] Post-process trace based on code validity and quality to obtain $\Lambda^{\mathrm{filter}}(\mathtt{T}^{\mathrm{single}}, \mathtt{C}^{\mathrm{single},*})$
15:     [Stage 3] Modify grids in trace via symbolic execution to obtain $\Lambda^{\mathrm{SE}}(\mathtt{T}^{\mathrm{single}}, \mathtt{C}^{\mathrm{single},*})$
16:     [Stage 4] Define $\Omega$ as the set of all $K'$-length subsequences of $\Lambda^{\mathrm{SE}}$
17:     [Stage 4] $\omega^* = \arg\min_{\omega \in \Omega} \mathcal{F}^{\Omega}_{\mathrm{complex}}(\omega; \mathtt{T}^{\mathrm{single}}, \mathtt{C}^{\mathrm{single},*}, K')$ as per Equation 2
18:     **Return** $\omega^* \in \Omega$

19: **function** PROGRESSYN$^{\mathrm{GRIDS}}$($\mathtt{T}^{\mathrm{ref}}, \mathtt{C}^{\mathrm{ref},*}, \sigma$)
20:     // When a fixed $\sigma$ is provided as input (the case when the procedure is invoked from PROGRESSYN)
21:     For $\sigma$, we define a sequence of $\mathtt{T}^{\mathrm{ref}}_{\mathrm{n}}$ tasks as $\omega^\sigma$
22:     We define the $k$-th task in $\omega^\sigma$ as follows:
23:         $\mathtt{C} \leftarrow \mathrm{REDCODE}(\{\mathtt{T}^{\mathrm{ref}}_{\mathrm{vis},\sigma_i}\}_{i=1,\ldots,k}; \mathtt{T}^{\mathrm{ref}}, \mathtt{C}^{\mathrm{ref},*})$
24:         $\mathtt{T} := (k, \{\mathtt{T}^{\mathrm{ref}}_{\mathrm{vis},\sigma_i}\}_{i=1,\ldots,k}, \mathtt{C}_{\mathrm{blocks}}, \mathtt{C}_{\mathrm{size}})$
25:         $k$-th subtask and solution code $:= (\mathtt{T}, \mathtt{C})$
26:     **Return** $\omega^\sigma$
27:     // When there is no $\sigma$ as input (the case when this procedure is used separately as discussed in Section 3.1)
28:     **Initialize:**  $\Sigma^{\mathtt{T}^{\mathrm{ref}}_{\mathrm{n}}} \leftarrow$ set of all permutations of $\{1, 2, \ldots, \mathtt{T}^{\mathrm{ref}}_{\mathrm{n}}\}$; $\Omega \leftarrow \{\}$
29:     **for** $\sigma \in \Sigma^{\mathtt{T}^{\mathrm{ref}}_{\mathrm{n}}}$ **do**
30:         For $\sigma$, we define a sequence of $\mathtt{T}^{\mathrm{ref}}_{\mathrm{n}}$ tasks as $\omega^\sigma$
31:         We define the $k$-th task in $\omega^\sigma$ as follows:
32:             $\mathtt{C} \leftarrow \mathrm{REDCODE}(\{\mathtt{T}^{\mathrm{ref}}_{\mathrm{vis},\sigma_i}\}_{i=1,\ldots,k}; \mathtt{T}^{\mathrm{ref}}, \mathtt{C}^{\mathrm{ref},*})$
33:             $\mathtt{T} := (k, \{\mathtt{T}^{\mathrm{ref}}_{\mathrm{vis},\sigma_i}\}_{i=1,\ldots,k}, \mathtt{C}_{\mathrm{blocks}}, \mathtt{C}_{\mathrm{size}})$
34:         $k$-th subtask and solution code $:= (\mathtt{T}, \mathtt{C})$
35:         Add $\omega^\sigma$ to $\Omega$
36:     $\omega^* = \arg\min_{\omega \in \Omega} \mathcal{F}^{\Omega}_{\mathrm{complex}}(\omega; \mathtt{T}^{\mathrm{ref}}, \mathtt{C}^{\mathrm{ref},*}, \mathtt{T}^{\mathrm{ref}}_{\mathrm{n}})$ as per Equation 2
37:     **Return** $\omega^* \in \Omega$

---

- PROGRESSYN$^{\mathrm{single}}$ is implemented in the function `progressyn_single()`
- PROGRESSYN$^{\mathrm{grids}}$ is implemented in the function `progressyn_grids()`

# D   Neural Program Synthesizers (Section 4.1)

In this section, we present further details on the evaluation of our subtasking algorithm with neural program synthesizers.

## D.1   Experimental Setup: Additional Details

Our neural program synthesizer is modelled based on the neural architecture from the work of Bunel et al. (2018). These neural models first embed each visual grid of the task using a CNN. Then, they run a decoder LSTM on the concatenation of previous code token in the synthesized code, and the embedding. The decoder output of all grids are maxpooled and masked by a syntax model. Our model architecture and hyperparameters are the same as in Bunel et al. (2018). As the syntax model for the codes, we used their provided hand-written syntax checker. We also followed their training strategy. Our reinforcement learning (RL) agent was initialized with pre-trained supervised network parameters. The same pretrained model was used for all the methods (PROGRESSYN, PROGRESSYN$^{\text{grids}}$, SAME-TC, and SAME-C). Afterwards, we used their vanilla RL training algorithm with different variants of the training dataset (as described in Section 4.1 and subsequently) and with expected rewards as the primary objective. Furthermore, we did not use beam search. We trained our models for 3.25 million gradient steps on a single 32GB Tesla V100S GPU and training took approximately 8 days.

For our experimental setup, we borrow the training dataset of 10000 *Karel programming tasks* from Bunel et al. (2018). During training, we further pruned the "small dataset of 10000 tasks" from Bunel et al. (2018) to obtain valid codes with full code coverage. Hence, our final training dataset comprised 7300 Karel tasks. Each training task in this set comprises of 6 visual grids. We generate variants of this dataset by augmenting it with subtasks synthesized for each training task using the following four methods: SAME-TC, SAME-C, PROGRESSYN$^{\text{grids}}$, and PROGRESSYN (as described in Section 4.1).

## D.2   Implementation Files

As our model of the neural program synthesizer is largely based on the work of Bunel et al. (2018), we borrowed their source code to run the experiments and adapted it slightly for reproducibility. It can be found under the folder "code/agents_neural/". This code base can be installed as a Python package. Specifically, the important commands of this package are defined in files under "code/agents_neural/cmds" and the key scripts are presented below:

- Train the model for a neural program synthesizer: `train_cmd.py`
- Evaluate the model for a neural program synthesizer: `eval_cmd.py`

We provide examples of how these commands can be run for our experiments in the files located under the folder "code/agents_neural/scripts/". For example, `finetune.sh` fine-tunes a pretrained model using reinforcement learning (RL).

## D.3   Relation to Execution-Guided Neural Program Synthesis (Chen et al., 2019)

Chen et al. (2019) proposes an execution-trace guided procedure for neural program synthesis (NPS) and is closely related to our approach for training NPS. Their approach dynamically breaks down the tasks into intermediate steps during a training episode and generates holes in the codes which are filled out. Our approach differs from this in two ways. First, their approach does not synthesize standalone subtasks, which is the focus of our study. Second, our approach to synthesizing subtasks is based on a "forward view" of the execution trace – intuitively this means that our algorithm generates subtasks by moving the goal of the task grid towards the avatar, and synthesizing solution codes. Chen et al. (2019)'s approach is based on a "backward view" of the execution trace – intuitively this means their approach moves the "avatar" of the task grid towards the "goal" and requires filling in holes in the solution codes.

### D.4 Limitations and Possible Extensions

In our current training setup for neural program synthesizers, we augment the training dataset with our synthesized subtasks. However, there may be more effective ways to incorporate the subtasks in the training process. For example, they could be used to explicitly design intermediate rewards which could yield similar performance gains with lower training time. Furthermore, in the current set up, we do not exploit the well-spaced code complexity of subtasks in the progression. However, the progression provides a natural curriculum for training these agents. Using such a curriculum could further improve their training time and performance.

## E  Search-based Agents (Section 4.2)

In this section, we analyze the utility of subtasks in guiding search-based agents to solve a specific block-based programming task, as a warm up to our study with novice human programmers (in Section 5). We model the search-based agents as Monte-Carlo learners and evaluate their performance on reference tasks $H_{08}$ and $H_{16}$. Next, we present details of our setup and discuss our results.

### E.1  Experimental Setup: Additional Details

At a high level, the Monte-Carlo agent is presented with a specific task $T^{ref}$. The goal of the agent is to navigate the space of codes to reach the solution code $C^{ref,*}$ via successive code edits, starting from empty code {RUN}. We guide the agent through the code space, via a progression of $K = 3$ subtasks for $T^{ref}$ to reach the solution code $C^{ref,*}$. Specifically, we divide the agent's search process into three stages, corresponding to the three subtasks of the progression. At each stage, we define the "goal code" as the solution of the corresponding subtask. We define the "starting code" as the code in which the agent begins the stage. In the first stage, the "starting code" is empty code {RUN}. In the following stages, it is updated to the "goal code" of the previous stage, if the agent successfully navigated to the "goal code".

Each stage is modeled as a deterministic finite-horizon Markov Decision Process(MDP) where states represent different (partial) codes and actions are possible edits in the code space (such as adding/deleting/updating a code block). Each episode starts at the "starting code" and terminates when the agent is in the "goal code" or the maximum number of steps, $H$, is reached. We set $H$ to be the minimum number of edits needed to reach the "goal code" from empty code (eg., for $H_{08}$ we set $H = 5$). We define our reward function as follows: 1 at the "goal code", and 0 in all other states. Each stage lasts for $N$ episodes. During the episodes, the agents learns the optimal tabular state-value function i.e., $V$-table. The agent follows an $\epsilon$-greedy policy and uses Monte-Carlo updates for its value function after each episode.

Furthermore, in order to capture our code complexity metric in an agent's search process (see Section 2), we bias the agent against picking actions that lead to states corresponding to codes with higher depths compared to their current state. This bias is applied only during tie-breaking. Specifically, we capture this bias using a softmax function with temperature parameter $\frac{1}{\beta}$ (henceforth, we refer to $\beta$ as the complexity bias parameter) i.e., the bias increases with increasing $\beta$ value. For example, from code {move} (depth = 1), an agent is more likely to pick an action that leads to code {move, move} (depth = 1) over an action that leads to code {REPEATUNTIL(goal){move}} (depth = 2).

For the five reference tasks used in the evaluation of our search-based agents (shown in Figure 5), we have the following hyperparameter configurations: $(N = 10000, H = 5, \beta = 7, \epsilon = 0.01)$ for $H_{08}$; $(N = 5000, H = 5, \beta = 7, \epsilon = 0.01)$ for $H_{12}$; $(N = 10000, H = 4, \beta = 7, \epsilon = 0.01)$ for $H_{16}$; $(N = 10000, H = 6, \beta = 7; \epsilon = 0.01)$ for $K_{Diag}$; $(N = 10000, H = 7, \beta = 7, \epsilon = 0.01)$ for $K_{sgStair}$.

With this set up, we trained 500 different agents, each with a different random seed. We ran the experiments on a machine with 3.30 GHz Intel Xeon CPU E5-2667 v2 processor and 256 GB RAM. Training a single agent for $N = 20,000$ episodes takes approximately 1.5 minutes. We parallelized the training of agents with different random seeds on 28 CPU cores, which takes approximately 25 minutes for 500 seeds.

## E.2 Methods Evaluated: Additional Details

We evaluate the performance of PROGRESSYN and three baseline methods which generate a progression of subtasks for a given $(\mathtt{T}^{\mathrm{ref}}, \mathtt{C}^{\mathrm{ref},*})$. The baseline methods are described in Section 4.2. For completeness, we mention them here again. First, we consider DEFAULT; this is the default setting where $K = 1$ and there is only one stage for the agent, corresponding to the reference task. Second, we consider methods SAME-TC and SAME-C (collectively referred to as SAME) which generate a progression of $K = 3$ subtasks which are not well-spaced w.r.t. their code complexity. Specifically, SAME-TC $= ((\mathtt{T}^k, \mathtt{C}^{k,*}))_{k=1,\dots,k}$ where $\forall k \in \{1,2,3\}, (\mathtt{T}^k, \mathtt{C}^{k,*}) = (\mathtt{T}^{\mathrm{ref}}, \mathtt{C}^{\mathrm{ref},*})$. SAME-C alters only the visual grids of subtasks in the progression while their solution codes remain the same as that of the reference task, i.e., $\forall k \in \{1,2,3\}, \mathtt{C}^{k,*} = \mathtt{C}^{\mathrm{ref},*}$. However, as our agents operate only in the code space, SAME-TC and SAME-C produce the same progression of subtasks, in this setting. Third, we consider methods CRAFTED-v1 and CRAFTED-v2 (collectively referred to as CRAFTED) which generate a handcrafted progression of $K = 3$ subtasks, that are well spaced w.r.t. their code complexity but have task grids that do not retain the visual context of the reference task. Figure 7 (in Section 5) and Figure 12 illustrate subtasks generated by these methods for reference tasks $\mathrm{H}_{16}$ and $\mathrm{H}_{08}$ respectively. Note that, in this setting the agents only use the solution codes of the progression as subtasks.

We additionally evaluate another baseline, PROGRESSYN$^{\mathrm{randomized}}$, where we randomize the order of subtasks synthesized by PROGRESSYN for each reference task. We evaluate the performance of our Monte-Carlo agents when presented with these subtasks and present the results in Appendix E.4.

## E.3 Validation: Additional Results

In this section, we discuss the importance of properties: task quality, task dissimilarity, task diversity, and well-spaced code complexity (as described in Section 2), for synthesizing a good progression of subtasks for a given reference task. In particular, properties of task quality, task dissimilarity and task diversity were added to our optimization problem (Equation 2) to ensure that we have a single optimal sequence of subtasks for a given reference task. These properties were specifically used for tie-breaking. We designed our baselines (SAME-TC, SAME-C, CRAFTED) in a manner that violated one or more desirable properties of the subtasks. Specifically,

- SAME-TC violates properties: (i) task diversity and (ii) well-spaced task complexity.
- SAME-C violates properties: (i) well-spaced task complexity because the solution codes of all the subtasks were the same.
- CRAFTED violates the following properties: (i) task dissimilarity w.r.t reference task.

So, the performance of these three baselines and PROGRESSYN with Monte-Carlo agents (discussed in Section 4.2) and novice human programmers (discussed in Section 5) highlights the degree to which these properties are important in achieving the overall goal of improving performance on the reference tasks.

We also provide the exact values of these properties for the progression of subtasks synthesized by each of the methods (DEFAULT, SAME, CRAFTED and PROGRESSYN) for reference tasks $\mathrm{H}_{08}$, $\mathrm{H}_{12}$, $\mathrm{H}_{16}$, $\mathrm{K}_{\mathrm{Diag}}$, and $\mathrm{K}_{\mathrm{sgStair}}$ in Figure 7. Specifically, in Figure 7 we present the following properties:

- Average Quality: This measures the average quality of subtasks in the final progression. In our implementation, we used the definition of task quality from Ahmed et al. (2020); in particular, we used a binary indicator of quality to be 1 if it is above a threshold.
- Normalized Task Diversity: We define the normalized task diversity in the final progression of subtasks as, $\frac{2}{K-1} \cdot \frac{\text{Task dissimilarity between subtasks}}{\text{Task dissimilarity between subtasks and the reference task}}$ where,
  - $K = $ Number of subtasks in the final progression
  - Task dissimilarity between subtasks $= \Sigma_{i=1,\dots,K} \Sigma_{j=1,\dots,i} \mathcal{F}_{\mathrm{diss}}^{\mathbb{T}}(\mathtt{T}^i, \mathtt{T}^j)$
  - Task dissimilarity between subtasks and the reference task $= \Sigma_{k=1,\dots,K} \mathcal{F}_{\mathrm{diss}}^{\mathbb{T}}(\mathtt{T}^k, \mathtt{T}^{\mathrm{ref}})$
- Maximum Complexity Jump: This measures the maximum difference in task complexity in the final progression of subtasks, where, task complexity is given by $\mathcal{F}_{\mathrm{complex}}^{\mathbb{T}}(\mathtt{T}) = 1000 * \mathtt{C}_{\mathrm{depth}}^{\mathtt{T},*} + \mathtt{C}_{\mathrm{size}}^{\mathtt{T},*}$.

| Method | Average Quality | | | | | | Normalized Task Diversity | | | | | | Maximum Complexity Jump | | | | | |
|---|---|---|---|---|---|---|---|---|---|---|---|---|---|---|---|---|---|---|
| | All | $H_{08}$ | $H_{16}$ | $H_{12}$ | $K_{Diag}$ | $K_{sgStair}$ | All | $H_{08}$ | $H_{16}$ | $H_{12}$ | $K_{Diag}$ | $K_{sgStair}$ | All | $H_{08}$ | $H_{16}$ | $H_{12}$ | $K_{Diag}$ | $K_{sgStair}$ |
| DEFAULT | 1. | 1. | 1. | 1. | 1. | 1. | 0. | 0. | 0. | 0. | 0. | 0. | 1405.4 | 1005 | 2004 | 1005 | 1006 | 2007 |
| SAME | 1. | 1. | 1. | 1. | 1. | 1. | 0.75 | 0.75 | 0.75 | 0.75 | 0.75 | 0.75 | 1405.4 | 1005 | 2004 | 1005 | 1006 | 2007 |
| CRAFTED | 1. | 1. | 1. | 1. | 1. | 1. | 1.21 | 1.1 | 1.13 | 1.31 | 0.75 | 1.75 | 1000. | 1002 | 1002 | 999 | 996 | 1001 |
| PROGRESSYN | 1. | 1. | 1. | 1. | 1. | 1. | 1.7 | 1.5 | 1.5 | 1.5 | 2. | 2. | 999.2 | 1000 | 1002 | 997 | 996 | 1001 |

Figure 7: Validation of metrics task quality, task dissimilarity, task diversity and task complexity for progression of subtasks synthesized by different algorithms. Higher values of average quality and normalized task diversity are better while lower values of maximum complexity jump are better; see details in Appendix E.3.

From Figure 7 we find that compared to all the baselines, PROGRESSYN achieves higher task diversity and has well-spaced task complexity (minimal value of maximum complexity jump in final progression of subtasks). Note that, baseline DEFAULT has only one task-code pair in the progression which is the same as the reference task and its solution code. Hence, we set its normalized diversity score to 0.

### E.4 Evaluation: Additional Results

**Effect of different hyperparameters.** We present the results of the performance of Monte-Carlo agents on reference tasks $H_{08}$ and $H_{16}$ in Figure 8. Specifically, we experimented with the following values of the hyperparameters $N \in \{5000, 10000, 20000\}$ (number of episodes in a stage); $\epsilon \in \{0, 0.01, 0.025\}$ (exploration factor); $\beta \in \{6, 7, 8\}$ (complexity bias parameter). For each hyperparameter setting, we used 500 agents, each generated from a different random seed. We report the fraction of agents (out of the 500 agents) that succeeded in solving reference tasks, guided by different subtasking methods. We find that DEFAULT performs significantly worse than all other methods, indicating the utility of subtasks in problem-solving. Furthermore, we find that, PROGRESSYN and CRAFTED perform significantly better than SAME in all hyperparameter settings. This indicates the utility of using progression of subtasks with well-spaced code complexity to guide problem-solving. Also, it can be seen that the success rate on $H_{08}$ is consistently higher than $H_{16}$ despite the latter requiring fewer edits. This shows that the $\beta$ parameter successfully captures our depth-focused complexity metric. Higher values of $\epsilon$ allows the agent to circumvent the complexity metric through unbiased exploration, which in return reduces the performance gap between $H_{08}$ and $H_{16}$. Finally, increasing $N$, increases the success rate of the agents because of longer training times.

**Comparison with ProgresSyn**[randomized]**.** We present the performance of the agents with methods DEFAULT, SAME, CRAFTED, PROGRESSYN and PROGRESSYN[randomized] on reference tasks $H_{08}$ and $H_{16}$ in Figure 9. Specifically, we compare PROGRESSYN and PROGRESSYN[randomized] and find that PROGRESSYN performs significantly better on both reference tasks. This indicates that well-spaced complexity in the sequence of subtasks is important for success on the reference task for our search-based agents.

### E.5 Implementation Files

The code files used for the experiments with the Monte-Carlo agents are in "code/agents_montecarlo/". The code graphs (space of partial codes starting with empty code {RUN}) for tasks $H_{08}$ and $H_{16}$ can be generated using the scripts in "code/agents_montecarlo/scripts". The source code for running Monte-Carlo agents on these graphs can be found in "code/agents_montecarlo/alg_rl". Specifically,

- Definition of Monte-Carlo agents: `mc_agent.py`
- Definition of the Markov Decision Process (MDP) for the agent: `env_graph_simple.py`
- Running the agent on the graph: `train_table.py`
- Evaluating the agent: `evaluate_setting_mp.py`

Further details can be found in "code/agents_montecarlo/README.md".

| Method | Fraction succeeded $N = 5,000$ | | | Fraction succeeded $N = 10,000$ | | | Fraction succeeded $N = 20,000$ | | |
|---|---|---|---|---|---|---|---|---|---|
| | Both | $H_{08}$ | $H_{16}$ | Both | $H_{08}$ | $H_{16}$ | Both | $H_{08}$ | $H_{16}$ |
| DEFAULT | 0.001 | 0.002 | 0.000 | 0.002 | 0.004 | 0.000 | 0.003 | 0.006 | 0.000 |
| SAME | 0.003 | 0.006 | 0.000 | 0.012 | 0.020 | 0.004 | 0.023 | 0.040 | 0.006 |
| CRAFTED | 0.589 | 0.944 | 0.234 | 0.711 | 0.996 | 0.426 | 0.857 | 1.000 | 0.714 |
| PROGRESSYN | 0.484 | 0.504 | 0.464 | 0.782 | 0.842 | 0.722 | 0.954 | 0.990 | 0.918 |

(a) Results for varying $N$ with $\epsilon = 0.01$ and $\beta = 7$

| Method | Fraction succeeded $\epsilon = 0$ | | | Fraction succeeded $\epsilon = 0.01$ | | | Fraction succeeded $\epsilon = 0.025$ | | |
|---|---|---|---|---|---|---|---|---|---|
| | Both | $H_{08}$ | $H_{16}$ | Both | $H_{08}$ | $H_{16}$ | Both | $H_{08}$ | $H_{16}$ |
| DEFAULT | 0.001 | 0.002 | 0.000 | 0.002 | 0.004 | 0.000 | 0.007 | 0.008 | 0.006 |
| SAME | 0.001 | 0.002 | 0.000 | 0.012 | 0.020 | 0.004 | 0.023 | 0.030 | 0.016 |
| CRAFTED | 0.331 | 0.616 | 0.045 | 0.711 | 0.996 | 0.426 | 0.869 | 1.000 | 0.739 |
| PROGRESSYN | 0.272 | 0.390 | 0.154 | 0.782 | 0.842 | 0.722 | 0.949 | 0.958 | 0.940 |

(b) Results for varying $\epsilon$ with $N = 10,000$ and $\beta = 7$

| Method | Fraction succeeded $\beta = 6$ | | | Fraction succeeded $\beta = 7$ | | | Fraction succeeded $\beta = 8$ | | |
|---|---|---|---|---|---|---|---|---|---|
| | Both | $H_{08}$ | $H_{16}$ | Both | $H_{08}$ | $H_{16}$ | Both | $H_{08}$ | $H_{16}$ |
| DEFAULT | 0.004 | 0.008 | 0.000 | 0.002 | 0.004 | 0.000 | 0.002 | 0.004 | 0.000 |
| SAME | 0.014 | 0.024 | 0.004 | 0.012 | 0.020 | 0.004 | 0.010 | 0.016 | 0.004 |
| CRAFTED | 0.768 | 1.000 | 0.536 | 0.711 | 0.996 | 0.426 | 0.687 | 0.984 | 0.390 |
| PROGRESSYN | 0.862 | 0.920 | 0.804 | 0.782 | 0.842 | 0.722 | 0.733 | 0.800 | 0.666 |

(c) Results for varying $\beta$ with $N = 10,000$ and $\epsilon = 0.01$

Figure 8: Results for Monte-Carlo learners with different hyperparameters and subtasking methods for reference tasks $H_{08}$ and $H_{16}$. We report the success rate (over 500 Monte-Carlo learners) on the reference task under column "Fraction succeeded" (higher scores being better). For all settings, we set the horizon length $H = 5$ for $H_{08}$ and $H = 4$ for $H_{16}$. The standard error of the success rate, for each combination of a task, a method and a hyperparameter setting, is less than 0.025. Note that, the second column of each subfigure has the same set of hyperparameters, $N = 10,000$, $\epsilon = 0.01$ and $\beta = 7$. We repeat the results for ease of comparison. Details of the experimental setup and results are presented in Appendix E.

| Method | Fraction succeeded $\beta = 6$ | | | Fraction succeeded $\beta = 7$ | | | Fraction succeeded $\beta = 8$ | | |
|---|---|---|---|---|---|---|---|---|---|
| | Both | $H_{08}$ | $H_{16}$ | Both | $H_{08}$ | $H_{16}$ | Both | $H_{08}$ | $H_{16}$ |
| DEFAULT | 0.004 | 0.008 | 0.000 | 0.002 | 0.004 | 0.000 | 0.002 | 0.004 | 0.000 |
| SAME | 0.014 | 0.024 | 0.004 | 0.012 | 0.020 | 0.004 | 0.010 | 0.016 | 0.004 |
| CRAFTED | 0.768 | 1.000 | 0.536 | 0.711 | 0.996 | 0.426 | 0.687 | 0.984 | 0.390 |
| PROGRESSYN$^{\text{randomized}}$ | 0.380 | 0.200 | 0.560 | 0.308 | 0.164 | 0.452 | 0.290 | 0.164 | 0.416 |
| PROGRESSYN | 0.862 | 0.920 | 0.804 | 0.782 | 0.842 | 0.722 | 0.733 | 0.800 | 0.666 |

Figure 9: Results for Monte-Carlo learners with hyperparameters $N = 10,000$, $\epsilon = 0.01$ and $\beta = 7$ and subtasking methods including PROGRESSYN$^{\text{randomized}}$ for reference tasks $H_{08}$ and $H_{16}$. We report the success rate (over 500 Monte-Carlo learners) on the reference task under column "Fraction succeeded" (higher scores being better). For all settings, we set the horizon length $H = 5$ for $H_{08}$ and $H = 4$ for $H_{16}$. The standard error of the success rate, for each combination of a task, a method and a hyperparameter setting, is less than 0.025. Details of the experimental setup and results are presented in Appendix E.

### E.6 Limitations and Possible Extensions

Next, we discuss a few limitations of our current study. In our current setup, the Monte-Carlo learners we use are limited by the fact that they operate only in the code space and don't consider the visual task grids. As a result, for reference task $H_{08}$, CRAFTED performs better than PROGRESSYN in all hyperparameter settings. This is because, the solution codes of subtasks in the progression generated by CRAFTED, are such that they progressively build towards the solution code of the reference task while requiring fewer code edits between successive subtasks. Comparatively, progressions generated by PROGRESSYN, while also building towards the solution code of the reference task, require higher number of code edits between successive subtasks (see CRAFTED and PROGRESSYN in Figure 12). A natural way to overcome this limitation is utilizing the visual grids during the training process of the Monte-Carlo agents. In fact, neural program synthesizers we used in our experiments in Section 4.2, which are conditioned on I/O pairs of the task, can be fine-tuned to solve a specific programming task. Another interesting extension of our current approach is exploring different ways to incorporate the generated subtasks in the training of Monte-Carlo learners. Our current setup was a warm up to our study with human programmers; however, Monte-Carlo learners in this setting are reduced to random search agents. Instead, our algorithm could also be utilized to explicitly design intermediate rewards based on the generated subtasks.

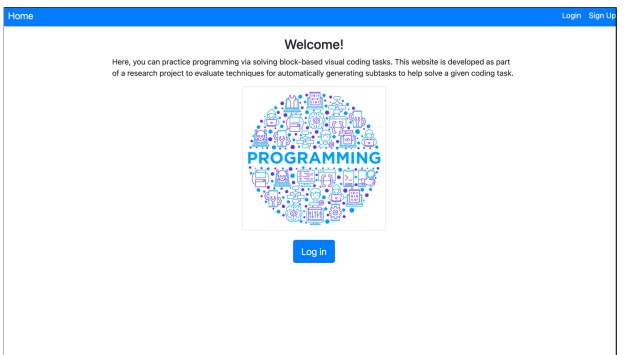

(a) Login and welcome page

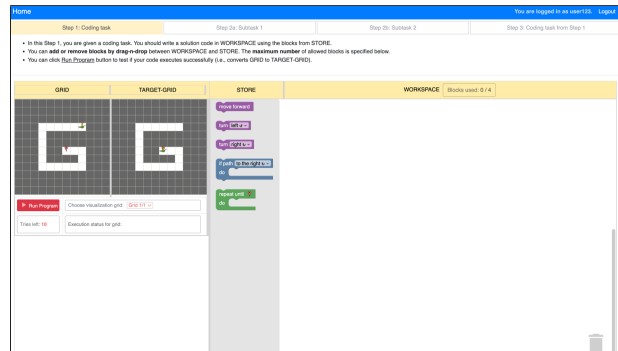

(b) Step 1: Introducing reference task $T^{ref}$

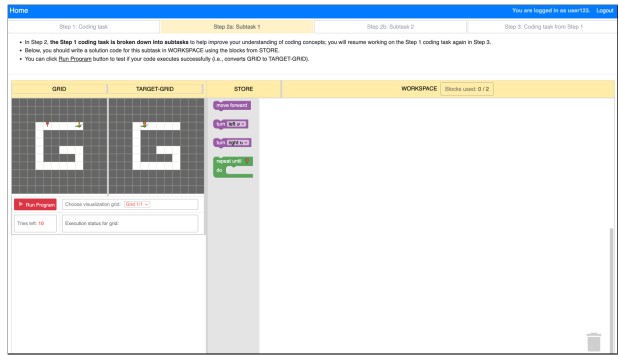

(c) Step 2: Solve progression of subtasks

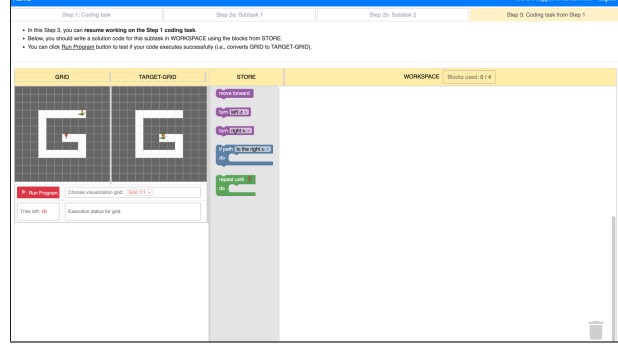

(d) Step 3: Solve $T^{ref}$

Figure 10: App Interface (See Appendix F and Footnote 5)

# F    Assisting Novice Human Programmers (Section 5)

In this section, we provide additional details about our study with novice human programmers. We begin by providing details about the web app used for the study, illustrate the subtasks synthesized by PROGRESSYN and other baseline methods for reference task $H_{08}$ and present additional analysis of the results obtained.

## F.1    App Interface: Additional Details

**Anonymous access to the app.** We illustrate the three stages of our web app in Figure 10.[5] On the app, we have enabled our subtasking algorithm PROGRESSYN and three block-based visual programming tasks: $H_{08}$, $H_{16}$, and *Karel programming environment* based STAIRWAY. A user can select one of these three tasks to practice on the platform, guided by a progression of subtasks synthesized by PROGRESSYN. Below, we present details to access our app *anonymously*. A user can go to the app link and login using any of the ten usernames listed. The passcode for all ten usernames is the same and is presented below.

- `Link`: https://www.teaching-blocks-subtasks.cc/
- `Username`: {reviewer1, reviewer2, reviewer3, reviewer4, reviewer5, reviewer6, reviewer7, reviewer8, reviewer9, reviewer10}
- `Passcode`: 000

---

[5]We illustrate an updated version of our app. In this version, a user can solve the reference task in Step 1. If successful, they exit the session. If unsuccessful, they proceed to Step 2 and solve the progression of subtasks.

### F.2 User Study Participants: Additional Details

Participants for the user study were recruited from Amazon Mechanical Turk. There were no potential participant risks anticipated for the study and an IRB approval had been obtained for the study. The participants were US-based adults, without expertise in block-based visual programming. The study took at most 30-35 minutes to complete, per participant. Each participant was remunerated with an amount of about 4 USD for every task they attempted.

### F.3 Methods Evaluated and Results: Additional Details

**Baseline methods on $H_{08}$.** As described in Section 5, we compare the performance of PROGRESSYN with baseline methods: DEFAULT, SAME-TC and SAME-C collectively referred to as SAME-C, CRAFTED-v1 and CRAFTED-v2 collectively referred to as CRAFTED. We evaluate the performance of the algorithms on reference tasks $H_{08}$ and $H_{16}$. Figure 7 in Section 5) and Figure 12 illustrate the progression of subtasks generated by each of these methods for reference tasks $H_{16}$ and $H_{08}$ respectively.

**Fine-grained results of baseline Same.** For our studies with novice human programmers, we also find a slight difference in the performance of baselines SAME-TC and SAME-C (which we together refer to as SAME). Figure 11 presents the performance of these baselines with human participants on the two reference tasks $H_{08}$ and $H_{16}$. Specifically, we find a slight improvement in performance of SAME-C compared to SAME-TC. This is because, the progression of subtasks synthesized by SAME-C have different visual task grids that are minimal modifications of the visual grid of the reference task, while having the same solution code as that of the reference task. However, in the progression of subtasks synthesized by SAME-TC, both the code and task grid are exactly the same as that of the reference task.

| Method | Total participants | | | Fraction succeeded | | |
|---|---|---|---|---|---|---|
| | Both | $H_{08}$ | $H_{16}$ | Both | $H_{08}$ | $H_{16}$ |
| SAME-TC | 114 | 57 | 57 | 0.667 | 0.842 | 0.491 |
| SAME-C | 116 | 59 | 57 | 0.672 | 0.847 | 0.491 |
| SAME | 230 | 116 | 114 | 0.669 | 0.845 | 0.491 |

Figure 11: The table shows results of the study with novice programmers on tasks $H_{08}$ and $H_{16}$. We present the fine-grained results of subtasking methods SAME-TC and SAME-C (collectively referred to as SAME). We report the success rate on different reference tasks under the column "Fraction succeeded" (higher scores being better); see Appendix F.3 for details.

### F.4 Limitations and Possible Extensions

Next, we discuss a few limitations of our current study. Our study was limited to under 600 participants given the high costs involved and the reported results are not statistically significant. However, a larger scale user study, with substantially more number of participants, would be needed to further validate the statistical significance of the results. It would also be interesting to conduct the study with the baseline PROGRESSYN$^{\text{randomized}}$, as we described for Monte-Carlo agents in Appendix E.2. Furthermore, we conducted our study with adult novice programmers. In the future it would be important to conduct longitudinal studies with real students to measure the pedagogical value of our algorithm. Finally, it would be interesting to evaluate extensions of our approach to more complex block-based programming tasks and domains.

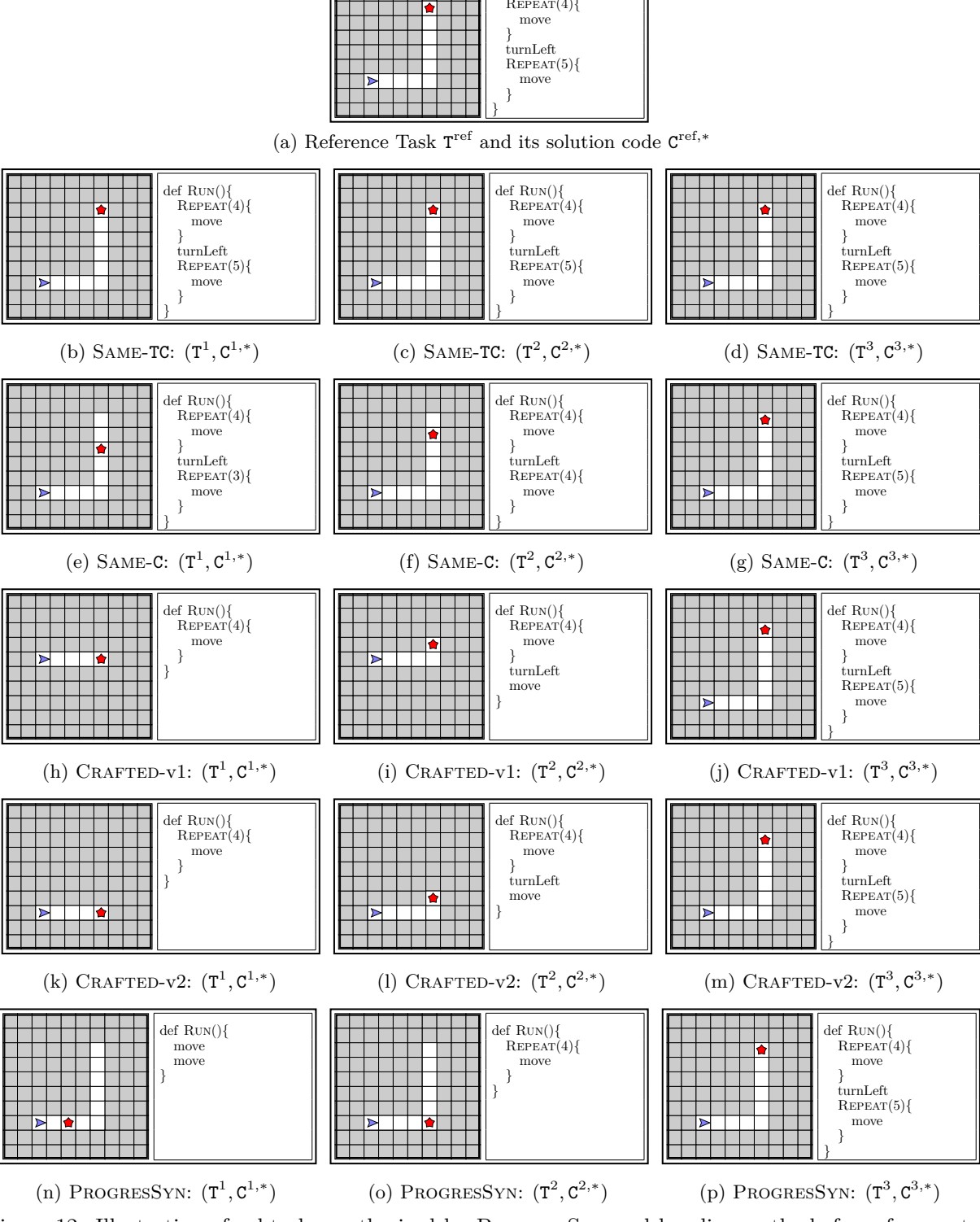

Figure 12: Illustration of subtasks synthesized by PROGRESSYN and baseline methods for reference task $H_{08}$. **(a)** shows reference task $H_{08}$. **(b)-(p)** shows subtasks generated by different methods. Each method synthesizes a progression of three subtasks, $((T^1, C^{1,*}), (T^2, C^{2,*}), (T^3, C^{3,*}))$. SAME-TC and SAME-C are collectively referred to as SAME; CRAFTED-v1 and CRAFTED-v2 are collectively referred to as CRAFTED. See Section 5, Appendix F for details.

