# OpenReview forum: "Synthesizing a Progression of Subtasks for Block-Based Visual Programming Tasks"
_TMLR — Rejected by TMLR_

### Review · Reviewer_xW8v · 2023-01-01

**Summary Of Contributions:**

Block-based programming is widely used in introductory computer science education and as a benchmark for neuro-symbolic AI. Block-based visual programming focuses on computational thinking and general problem-solving by reducing the burden of learning syntax. This aspect is also useful in evaluating neuro-symbolic AI where the focus is on AI's logical reasoning and problem-solving capability. The paper proposes algorithms that can synthesize a progression of subtasks for block-based visual programming tasks. The synthesized subtasks are high-quality, well-spaced in terms of their complexity, and solving this progression leads to solving the reference task. The developed approach is demonstrated on tasks in the Karel programming environment (Pattis et al., 1995). A user study is used to demonstrate that the synthesized progression of subtasks can assist a novice programmer in solving tasks in the Hour of Code: Maze Challenge.

**Audience:**

Yes

**Broader Impact Concerns:**

There are no broader impact concerns with the paper.

**Claims And Evidence:**

Yes

**Requested Changes:**

For a relatively simpler idea, the notation is overly complicated. For example, why use a mixture of superscript and subscript in T_vis,i and T^task_id ? The reviewer recommends putting a table with symbols in the paper to make the paper easy to read if simplifying the notation would require too much of rewrite.

In the definition of omega, don't we need T^k_n < T^(k+1)_n. Also, what does subscript n denote? T_n is the number of visual grids.

There are also some questions in the strengths/weaknesses questions that need clarification.



**Strengths And Weaknesses:**

Strengths

- The paper addresses an important problem of decomposing tasks and synthesizing solutions for it.

-  User study was conducted to study the usefulness of subtasking and the well-spacing of code complexity.


Weaknesses

- While the approach is presented as a general technique for decomposing tasks into subtasks, its description is very tightly connected to the simplistic grid world examples. The dissimilarity uses Hamming distance that would not generalize to other problems. The complexity of code using AST is also a gross over-simplification - treating each node in the AST as being homogeneous. The complexity of a task is defined as the complexity of the simplest code solving the task. Is it practical to expect being able to find all code that solves a task (once the T_store becomes large enough)? A good conceptual definition should also be computable. This is particularly important because complexity of subtasks are an important part of the overall approach.

-  ProgresSynsingle appears to use the solution Cref (its partial traces) to find the subtasks. This appears a bit cyclic approach that would be sensitive to the solution Cref. For the problem of grid world, this might not be that critical but typically, a number of possible solutions exist for a problem and making the problem decomposition depend on the solution can make the decomposition quality heavily dependent on Cref. Is the reviewer missing something?

- ProgresSyn relies on exhaustive enumeration and  ProgresSynGrid - does this mean the approach in the paper is limited to only grid problems?

Overall, the reviewer is positive about the paper, but some of the main contributions need further clarification.

---

> ### Author Response · Authors · 2023-01-27
> **Response to Reviewer xW8v**
>
> Thank you for carefully reviewing our paper! We greatly appreciate your feedback. Please see below our responses to your comments. We will upload a revised version of the paper in the next few days.
>
> ---
>
> **1. Regarding tractability of the task complexity definition and generalizability of the dissimilarity/complexity metrics to other domains.**
>
> - Our framework for generating subtasks is designed for the domain of visual block-based programming. In this domain, the space of solution codes for a task (constrained by the number of blocks used) is typically finite and tractable. Hence, for this domain, it is possible to find the complexity of the simplest code that solves the task.
> - However, as the reviewer rightly pointed out, our Hamming distance-based dissimilarity metric will not generalize to other programming domains. While our framework cannot be directly applied to other domains (e.g., text-based programming languages and tasks), some of the ideas presented could potentially be extended if the following elements are redesigned: (a) task synthesis for programs in the domain using symbolic execution; (b) task complexity; (c) task dissimilarity. This extension to other domains is an interesting direction for future work.
>
>
> ---
>
> **2. Regarding the role of $C_{ref}$ in the algorithm.**
>
> The reviewer rightly pointed out that the quality of our decomposition framework is heavily dependent on $C^{ref}$. When the reference task has multiple solution codes (i.e., multiple $C^{ref}$), our framework can be easily extended to obtain a progression of subtasks by considering multiple $C^{ref}$ to obtain the set $\Omega$, and then solving Equation 2 (Section 2.2) to obtain the optimal progression of subtasks.
>
> ---
>
> **3. ProgresSyn relies on exhaustive enumeration and ProgresSynGrid - does this mean the approach in the paper is limited to only grid problems?**
>
> We have provided a discussion on this point in the appendix (titled “$\text{ProgresSyn}^\text{grids}$: Additional Details.” ). More concretely:
> - When the number of visual grids $T^{ref}_{n}$ of a task $T^{ref}$ is large, we can use a more tractable strategy to fix a sequence of grids instead of an exhaustive enumeration of all possible sequences of grids. For example, we can use a greedy strategy to fix a sequence of grids based on the degree of code coverage w.r.t. the solution code. This greedy strategy would begin with a grid corresponding to the maximum code coverage of the solution code and sequentially add the remaining grids.
> - When the number of visual grids is small, we can rely on exhaustive enumeration to optimize for Equation 2 and obtain the progression.
>
> ---
>
> **4. Regarding the simplification of notation in the paper.**
>
> We thank the reviewer for their suggestion. We will add a detailed table of notations in the appendix.
>
> ---
>
> **5. In the definition of omega, don't we need T^k_n < T^(k+1)_n. Also, what does subscript n denote? T_n is the number of visual grids.**
> - In our setup, the subscript $n$ is an identifier for the number of grids. Specifically, for a given task $T$, $T_n$ represents the total number of visual grids of the task.
> -  Furthermore, $\Omega$ is defined as the collection of all possible progression of subtasks for a reference task (Section 2.2). So, we do not explicitly impose the constraint that $T^{k}_n < T^{k+1}_n$ in the definition of $\Omega$. The constraint $T^{k}_n \leq T^{k+1}_n$ is a property of the progression of subtasks synthesized by our algorithm $\text{ProgresSyn}$, and is not a part of the definition of $\Omega$.
>
> ---
> We hope that our responses address your concerns. We will also upload a revision in the next few days. If you have any other comments or feedback, please let us know! We will be happy to provide further responses. We are looking forward to hearing back from you! Thank you again for the review.

---

> ### Author Response · Authors · 2023-01-31
> **Response to Reviewer xW8v - Paper Revision**
>
> Dear Reviewer xW8v,
>
> We have now uploaded a revised version of the paper. We have highlighted the changes in blue color text. In this revision, we have incorporated the feedback in the main paper and appendices. Please let us know if you have any other comments or feedback.
>
> Thank you!
>
> Authors

---

### Review · Reviewer_1gq7 · 2023-01-05

**Summary Of Contributions:**

This paper proposes ProgresSyn, an algorithm to generate a progression of subtasks given a visual programming task from input-output pairs. The increasingly complicated subtasks can provide more fine-grained guidance that leads human programmers and neural program synthesizers to better solve a challenging programming task. The ProgresSyn algorithm first executes the program to obtain the execution trace for each input grid. Afterward, the algorithm post-processes the execution trace to obtain potential solution codes of subtasks, then generates subtasks for a single grid. Finally, the algorithm combines subtasks for different subsets of grids.

They evaluate ProgresSyn on the Karel domain. First, they use an existing neural program synthesizer as the base model, and show that augmenting the training data with ProgresSyn improves the performance. Afterward, they evaluate a Monte-Carlo agent and conduct a user study with human programmers, and in both experiments, they show that using subtasks generated by ProgresSyn improves the success rate of the agent and human programmers.

**Audience:**

Yes

**Claims And Evidence:**

No

**Requested Changes:**

1. Use more programming tasks for the evaluation of the Monte-Carlo agent and human programmers. Also, include an analysis of the performance on tasks with different types of solution code.

2. Present the generalization accuracy for all experiments, instead of only the exact match accuracy.

3. Explain why Same outperforms Default and Crafted in the user study.

3. Modify the Monte-Carlo agent to use an encoder for input visual grids, similar to the one in the neural program synthesizer.

4. Explain why the neural program synthesizer performance is much worse than results in Bunel et al.

5. Compare ProgresSyn to other approaches that train the neural program synthesizer with a larger training data, to justify its effectiveness.

**Strengths And Weaknesses:**

Strength: Generating a progression of subtasks to solve a more complicated task is a good topic, both in program synthesis and other domains. In particular, curriculum design is important for programming education.

Weaknesses: While I think the submission studies an important topic, I don't think the evaluation is convincing enough.

1. Testing only on 2 tasks for the Monte-Carlo agent and human programmers is insufficient. As shown in the results, there is a high variance between the performance on 2 tasks, and we can have different conclusions when comparing different subtask generation algorithms on different tasks. Also, there is no good justification on why the authors choose those 2 tasks. In general, the evaluation should be conducted on more programming tasks of different complexity, and there should be an analysis of the performance of tasks with different types of solution code.

2. In general, I think the generalization accuracy is a better evaluation metric than the exact match accuracy. The authors only present the generalization accuracy in the supplementary material for the neural program synthesizer experiment. I suggest the authors to present the generalization accuracy for all experiments.

3. In the user study, I do not understand why Same outperforms Default and Crafted, since all subtasks in Same have the same ground truth code as the reference task.

3. For the Monte-Carlo agent, it is strange that the agent does not use the visual input to guide the search. As a result, Crafted can be a better algorithm even if the visual grids of subtasks are very different from the reference task. It is more reasonable to use an encoder like the one in the neural program synthesizer.

4. The evaluation of the neural program synthesizer is also problematic. First, from the supplementary material, I suppose they use Bunel et al.'s model trained with RL, which should already achieve >17% exact match accuracy and >25% generalization accuracy with the 10K training data, based on the original paper. On the other hand, ProgresSyn achieves a similar accuracy, while using 9x training data. Also, note that the full training dataset from Bunel et al. has >1M samples. From [1], we can observe that using a larger subset from the full training data has much better performance than using ProgresSyn for data augmentation. Therefore, more work needs to be done to justify whether the ProgresSyn is helpful for training.

[1] Xinyun Chen, Dawn Song, Yuandong Tian, Latent Execution for Neural Program Synthesis, NeurIPS 2021.

---

> ### Author Response · Authors · 2023-01-27
> **Response to Reviewer 1gq7 - Part 1**
>
> Thank you for carefully reviewing our paper! We greatly appreciate your feedback. Please see below our responses to your comments. We will upload a revised version of the paper in the next few days.
>
> ---
>
> **1. Use more programming tasks for the evaluation of the Monte-Carlo agent and human programmers. Also, include an analysis of the performance on tasks with different types of solution code.**
>
> - Evaluation with human learners: We evaluated the performance of our subtasking framework with human learners on only two tasks due to the high cost involved as mentioned in the appendix (titled “Assisting Novice Human Programmers”). More concretely, the total cost of conducting the user study for two tasks was about 2500 USD and we had to restrict the number of tasks. We did ensure that the two tasks used in our study significantly differ in their task complexity (one task using two Repeat blocks; another task using repeatUntil with nested If).
> - Evaluation with Monte-Carlo agents: As suggested by the reviewer, we evaluated our algorithm on more diverse tasks with Monte-Carlo agents. Below, we present the additional results. We will add these results to the appendix.
>
> *Performance: Fraction of Monte-Carlo agents successful in solving the reference task*
> | Method | ALL | $H_{08}$ | $H_{16}$ | $H_{12}$ | $K_{Diag}$ | $K_{Stair}$ |
> |---|---|---|---|---|---|---|
> | $\text{Default}$ | $0.018$ | $0.004$ | $0.000$ | $0.084$ | $0.000$ | $0.000$ |
> | $\text{Same}$ | $0.055$ | $0.020$ | $0.004$ | $0.238$ | $0.012$ | $0.000$ |
> | $\text{Crafted}$ | $0.783$ | $0.996$ | $0.426$ | $0.964$ | $0.946$ | $0.628$ |
> | $\text{ProgresSyn}$ | $0.836$ | $0.842$ | $0.722$ | $0.998$ | $0.992$ | $0.628$ |
>
> *Task diversity: Properties of the reference tasks*
>
> | Property | $H_{08}$ |  $H_{16}$ | $H_{12}$ | $K_{Diag}$ | $K_{Stair}$ |
> |---|---|---|---|---|---|
> | Task type | HOC |  HOC |  HOC |  Karel |  Karel |
> | Task complexity | $1005$ |  $2004$ |  $1005$ |  $1006$ | $2007$ |
> | Depth of solution code ($C_\text{depth}$) | $1$ |  $2$ |  $1$ |  $1$ |  $2$ |
> | Size of solution code ($C_\text{size}$) | $5$ |  $4$ |  $5$ |  $6$ |  $7$ |
>
> ---
>
> **2. Present the generalization accuracy for all experiments, instead of only the exact match accuracy.**
>
> We thank the reviewer for their suggestion. We will update the training plot in Figure 4a with the generalized accuracy metric.
>
> ---
>
> **3. Explain why Same outperforms Default and Crafted in the user study.**
>
> - $\text{Same}$ vs. $\text{Default}$: “$\text{Default}$” refers to the method where human learners are directly presented with the reference task without any subtasks. “$\text{Same}$” on the other hand, has subtasks that are all either exactly the same as the reference task, or have the same solution code as it. So, in the algorithm “$\text{Same}$”, learners get more attempts to solve the reference task than “$\text{Default}$” (i.e., 40 attempts in comparison to 10 attempts). Hence, “$\text{Same}$” outperforms “$\text{Default}$”.
> - $\text{Same}$ vs. $\text{Crafted}$: We find that “$\text{Same}$” outperforms “$\text{Crafted}$”, although the difference in performance between these methods is not significant. We hypothesize that this is the case because “$\text{Crafted}$” synthesizes subtasks that are visually very different from the reference task. This could possibly make the progression of subtasks more confusing w.r.t the reference task, leading to lower success on the reference task.
>
> ---
>
> **4. Modify the Monte-Carlo agent to use an encoder for input visual grids, similar to the one in the neural program synthesizer.**
>
> We thank the reviewer for suggesting this direction to incorporate visual grids with Monte-Carlo agents. There seem to be several non-trivial challenges in building these agents. Below, we highlight some thoughts on this direction and the challenges involved.
> - One possible approach is to use the NPS architecture as a basis for the Monte-Carlo agent. However, a key challenge with this approach is that the NPS architecture used in our work builds the code sequentially, whereas Monte-Carlo agents should be able to make non-trivial edits (e.g., nesting existing blocks within a while block) in their codes to leverage the benefits of subtasks in a progression.
> - Another possible approach is to extend our current implementation by incorporating visual grid encoding as part of the state space representation (i.e., the edit-graph in our case); however, this would require designing a new neural architecture that can learn over such a graph.
>
> ---
> (the response is continued in Part 2)

---

> ### Author Response · Authors · 2023-01-28
> **Response to Reviewer 1gq7 - Part 2**
>
> (continuation of the response from Part 1)
>
> ---
>
> **5. Explain why the neural program synthesizer performance is much worse than results in Bunel et al.**
>
> For the evaluation of our framework with neural program synthesizers, our experimental setup differs from the work of [Bunel et al. 2018] in the following ways:
> - The main difference is in the training dataset. We noticed that the original dataset from [Bunel et al. 2018] has tasks where the solution code doesn’t have full coverage. As we focused our framework on good quality tasks where the solution code has full coverage, we pruned tasks from the “small dataset of 10,000 tasks” from [Bunel et al. 2018] where the solution code didn’t have full coverage w.r.t. the task (e.g., redundant blocks). Hence, the final training dataset used in our experiments comprised 7,300 Karel tasks.
> - In the training process, we focused on their RL-based training component, starting from the same pre-trained model using the supervised learning component (see discussion of the experimental setup in the appendix titled “Neural Program Synthesizers” ). As our goal was to evaluate the helpfulness of subtasks in learning to solve a given reference task, the RL-based training component is more suitable to evaluate our subtasking framework (as it implicitly helps in dealing with exploration). However, this RL training component is much slower than the supervised learning component (about 8 days for 3.25 million gradient steps). To speed up the training process, we did not use beam search during RL training in contrast to [Bunel et al. 2018] where beam search is used during training.
>
> [Bunel et al. 2018] Leveraging Grammar and Reinforcement Learning for Neural Program Synthesis.
>
> ---
>
> **6. Compare ProgresSyn to other approaches that train the neural program synthesizer with a larger training data, to justify its effectiveness.**
>
> In our training process for NPS, we did not use the larger 1M training dataset from [Bunel et al. 2018] because of the following reasons:
> - As discussed in our response to point 5., training the RL component of NPS agents is much slower than the supervised learning component. Specifically, our training process on the “small dataset of 10,000 tasks” took 8 days for 3.25 million gradient steps. As a result, we were unable to scale the training process to larger training datasets.
> - Furthermore, methods that directly use a larger training dataset are not directly comparable with our method because our method augments the given training data by synthesizing subtasks from the existing tasks. Apart from the data augmentation baselines in the paper, we have now included an additional baseline in our evaluation, namely $\text{Same-C}$. We report results below and will also include them in the updated paper.
>
> $\text{Same-C}$ baseline augments the training dataset with subtasks synthesized by $\text{Same-C}$ as described in Section 4.2. More concretely, for each training task in the dataset, we generate a progression of subtasks by only altering the visual grid of the reference task while the solution code remains the same as that of the reference task. The number of subtasks generated by this method for each task in the dataset is the same as that generated by $\text{ProgresSyn}$ (i.e., 9 subtasks in the progression for a given task). We present the final performance of these baselines along with the existing methods below:
>
> *Final performance: Exact and Generalized accuracy metric for NPS agents trained using different data augmentation methods*
>
> | Method | Exact | Generalized |
> |---|---|---|
> | $\text{Same-TC}$ | $10.6 \pm 0.1$ |  $14.7 \pm 0.2$ |
> | $\text{ProgresSyn}^{\text{grids}}$ | $13.3 \pm 0.3$ |  $18.8 \pm 0.5$ |
> | $\text{Same-C}$ | $13.7 \pm 0.2$ |  $19.2 \pm 0.3$ |
> | $\text{ProgresSyn}$ | $17.3 \pm 0.3$ |  $24.6 \pm 0.5$ |
>
>
>
> *Training performance: Generalized accuracy of neural program synthesizers by gradient step, trained using different data augmentation methods*
>
> | Method | Step 0.5M | Step 1M | Step 1.5M | Step 2M | Step 2.5M | Step 3M |
> |---|---|---|---|---|---|---|
> | $\text{Same-TC}$ | $13.76$ | $14.10$ | $14.12$ | $13.58$ | $13.83$ | $13.02$ |
> | $\text{ProgresSyn}^{\text{grids}}$ | $15.72$ | $15.49$ | $16.07$ | $16.79$ | $18.44$ | $18.75$ |
> | $\text{Same-C}$ | $15.92$ | $16.29$ | $17.24$ | $18.05$ | $17.70$ | $18.69$ |
> | $\text{ProgresSyn}$ | $17.50$ | $20.25$ | $21.40$ | $21.16$ | $23.10$ | $23.32$ |
>
> ---
> We hope that our responses address your concerns. We will also upload a revision in the next few days. If you have any other comments or feedback, please let us know! We will be happy to provide further responses. We are looking forward to hearing back from you! Thank you again for the review.

---

> ### Author Response · Authors · 2023-01-31
> **Response to Reviewer 1gq7 - Paper Revision**
>
> Dear Reviewer 1gq7,
>
> We have now uploaded a revised version of the paper. We have highlighted the changes in blue color text. In this revision, we have incorporated the feedback in the main paper and appendices. Please let us know if you have any other comments or feedback.
>
> Thank you!
>
> Authors

---

### Review · Reviewer_94Ar · 2023-01-16

**Summary Of Contributions:**

This paper proposes to solve the progression generation of sub-tasks for block-based visual programming tasks. Specifically the paper first defines the visual programming tasks, and then define the progression generation through a set of metrics, including:
- the code complexity in terms of the size/depth;
- the difficulty of each task, defined by the minimum code complexity;
- and several other metrics.

The goal is to synthesize a sequence of subtasks, such that the discrepancy of task difficulties is minimized (i.e., trying to seek for a smooth transition between tasks).
Experiments are done on both synthetic tasks in the Karel programming environment, as well as real-world user studies. The paper has shown that the proposed method is able to effectively improve the task solving through progression.


**Audience:**

Yes

**Claims And Evidence:**

Yes

**Requested Changes:**

Corresponding to the above potential weaknesses, the paper might be better if one can address these issues. Specifically:
- try to simplify the notation a bit more, by using algorithm boxes, illustrations, or deferring unnecessary details into appendix;
- have a proper definition of ‘helpfulness’ between tasks, i.e., why solving a certain task would increase the chance of solving another task.
- try to connect the progression generation to a broader set of tasks, and briefly mention how one can generalize the proposed approach.


**Strengths And Weaknesses:**

Strength:
- The paper tackles an interesting problem of progression generation.
- The authors have tried their best to come up with the proper definition of their progression, which I think is relatively comprehensive and thorough.
- The experiments on both synthetic and user studies are impressive.

Weakness:
- The notation is a bit heavy, which makes it difficult to parse in some parts. For example instead of defining permutation in Sec 3.1, could it be easier to get rid of this definition and use something equivalent (maybe with another algorithm box)?
- The scope of this paper is somewhat limited. I’m not sure if focusing solely on block-based visual programming is interesting enough for a broad set of people. The techniques behind the paper are somewhat tied to the task, which might be limiting the generalization of the notion of progression generation. Actually the part that I like most is the progression generation, hopefully on more general domains.
- The current definition of sub-tasks might have some corner cases that can be tricky. For example, if smoothness is the major objective, what if someone comes up with a sequence of subtasks that have smooth growth in terms of difficulty, but have no correlation with the final task? I guess in this paper the specific domain which is the visual block synthesis might offer the restriction of task similarities, but in general it is not sure how to come up with a sequence of sub-tasks that can be measured to be *helpful* for the final task.

---

> ### Author Response · Authors · 2023-01-27
> **Response to Reviewer 94Ar**
>
> Thank you for carefully reviewing our paper! We greatly appreciate your feedback. Please see below our responses to your comments. We will upload a revised version of the paper in the next few days.
>
> ---
>
>
> **1. Regarding the use of heavy notation in the paper.**
>
> We thank the reviewer for the feedback. To improve readability, we will add a detailed table of notations in the appendix.
>
> ---
>
> **2. Regarding the “helpfulness” between tasks, i.e., why solving a certain task would increase the chance of solving another task.**
>
> - In our framework, we design a progression of subtasks *specifically* for the given reference task. The goal is that solving this progression of subtasks would lead to an increase in the success rate of solving the reference task. We evaluate the *helpfulness* of the synthesized progression by measuring the increase in success rate when the reference task is solved along with the progression of subtasks. Empirically (with agents and human participants), we find that solving the progression of subtasks improves the success rate on the reference task (Sections 4 and 5).
> - We would like to clarify that our algorithm, by design, ensures the correlation of the progression of subtasks with the reference task. More specifically, it generates solution codes for subtasks which are reductions of the solution code of the reference task. Moreover, we pick a small sequence of subtasks by optimizing for smoothness in terms of the complexity of subtasks in the generated progression.
>
> ---
>
> **3. Regarding the scope of the paper and generalization of our proposed approach to other domains.**
>
> Our current framework is designed for block-based visual programming environments, given its recent popularity in CS Education as well as in benchmarking neural synthesis techniques. While the proposed algorithm itself is not directly applicable to other domains (e.g., text-based programming languages), the key ideas presented could potentially be extended by redesigning the following elements in our framework: (a) task synthesis for programs in the domain using symbolic execution; (b) task complexity; (c) task dissimilarity. This extension to other domains is an interesting direction for future work.
>
>
> ---
> We hope that our responses address your concerns. We will also upload a revision in the next few days. If you have any other comments or feedback, please let us know! We will be happy to provide further responses. We are looking forward to hearing back from you! Thank you again for the review.

---

> ### Author Response · Authors · 2023-01-31
> **Response to Reviewer 94Ar - Paper Revision**
>
> Dear Reviewer 94Ar,
>
> We have now uploaded a revised version of the paper. We have highlighted the changes in blue color text. In this revision, we have incorporated the feedback in the main paper and appendices. Please let us know if you have any other comments or feedback.
>
> Thank you!
>
> Authors

---

### Decision · Action_Editors · 2023-03-05

**Recommendation:** Reject

**Comment:**

The reviewer opinions were mostly on the reject side, with 2 leaning reject and 1 weak support for acceptance.  The concerns were mostly on the narrow scope, limited audience, and evaluation which one reviewer commented as not convincing enough.  Based on the reviews and my comments above for claims & evidence and audience, this paper is not yet at a sufficient level for publication at TMLR, with the narrow scope being the main issue limiting its audience and impact.

If the authors can demonstrate the effectiveness of the proposed method on other programming tasks outside the current visual grid based settings, the work could be more impactful and interesting to a larger audience, in which case the authors should consider submitting again.

**Audience:**

This paper targets the specific problem of generating a progression of subtasks for block-based visual programming problems.  The topic of creating a progression of subtasks for solving a challenging problem has its importance and could be quite valuable for educational use cases.  However the specific setup used in this paper is relatively narrow and restricted to the particular type of visual grid programming tasks, making its scope quite a bit narrower than it could be.  The proposed algorithm also would have problem generalizing to other programming tasks.

This is the main issue for this paper and the concern on scope of the paper was raised by all reviewers.

**Claims And Evidence:**

The authors propose a method to generate a progression of subtasks for guiding learners to solve visual programming tasks.  The proposed approach does produce a progression of subtasks optimizing a certain objective.

The main claims from the authors are:
1. They formalize an objective of synthesizing a progression of subtasks for block-based visual programming tasks and present a novel algorithm to optimize this objective, which was well supported by the content of the paper.
2. They showcase their proposed algorithm towards assisting problem solving performance of AI agents as well as novice programmers.  This claim is the one that the reviewers and myself have some concerns about.
3. They will share the web app for future study, which I believe the authors would do as they have included relevant resources in the supplementary material.

Regarding claim #2, there are two main issues:
1. The authors did not have the resources to train a model for the Karel task on the full dataset with 1M examples as one reviewer pointed out, making the base model quite a bit worse than state-of-the-art (from a few years ago).  The results in this setting is therefore less convincing.
2. Personally I think the human evaluation setup is a bit questionable - giving people hints on a problem should be intuitively helpful for solving that very problem, which is what the experiment tested.  But in fact, a “cheating” setup where the hints directly give away the final solution to the reference task would probably achieve even higher metrics.  Hence getting a higher score in this setting might not be that useful.  Perhaps what’s more important (up for debate) is how those hints on the training problems help the learners when they face new problems that they need to solve without seeing the additional hints.